# Polynucleotides Suppress Inflammation and Stimulate Matrix Synthesis in an In Vitro Cell-Based Osteoarthritis Model

**DOI:** 10.3390/ijms241512282

**Published:** 2023-07-31

**Authors:** Sree Samanvitha Kuppa, Hyung-Keun Kim, Ju-Yeon Kang, Seok-Cheol Lee, Hong-Yeol Yang, Jaishree Sankaranarayanan, Jong-Keun Seon

**Affiliations:** 1Department of Biomedical Sciences, Chonnam National University Medical School, Hwasun 58128, Republic of Korea; 2Department of Orthopaedics Surgery, Center for Joint Disease of Chonnam National University Hwasun Hospital, 322 Seoyang-ro, Hwasun-eup 519-763, Republic of Korea; 3Korea Biomedical Materials and Devices Innovation Research Center, Chonnam National University Hospital, 42, Jebong-ro, Dong-gu, Gwangju 501-757, Republic of Korea

**Keywords:** osteoarthritis, chondrocytes, inflammation, hypertrophy, extracellular matrix, polynucleotide

## Abstract

Osteoarthritis (OA) is characterized by degeneration of the joint cartilage, inflammation, and a change in the chondrocyte phenotype. Inflammation also promotes cell hypertrophy in human articular chondrocytes (HC-a) by activating the NF-κB pathway. Chondrocyte hypertrophy and inflammation promote extracellular matrix degradation (ECM). Chondrocytes depend on Smad signaling to control and regulate cell hypertrophy as well as to maintain the ECM. The involvement of these two pathways is crucial for preserving the homeostasis of articular cartilage. In recent years, Polynucleotides Highly Purified Technology (PN-HPT) has emerged as a promising area of research for the treatment of OA. PN-HPT involves the use of polynucleotide-based agents with controlled natural origins and high purification levels. In this study, we focused on evaluating the efficacy of a specific polynucleotide sodium agent, known as CONJURAN, which is derived from fish sperm. Polynucleotides (PN), which are physiologically present in the matrix and function as water-soluble nucleic acids with a gel-like property, have been used to treat patients with OA. However, the specific mechanisms underlying the effect remain unclear. Therefore, we investigated the effect of PN in an OA cell model in which HC-a cells were stimulated with interleukin−1β (IL−1β) with or without PN treatment. The CCK-8 assay was used to assess the cytotoxic effects of PN. Furthermore, the enzyme-linked immunosorbent assay was utilized to detect MMP13 levels, and the nitric oxide assay was utilized to determine the effect of PN on inflammation. The anti-inflammatory effects of PN and related mechanisms were investigated using quantitative PCR, Western blot analysis, and immunofluorescence to examine and analyze relative markers. PN inhibited IL−1β induced destruction of genes and proteins by downregulating the expression of MMP3, MMP13, iNOS, and COX-2 while increasing the expression of aggrecan (ACAN) and collagen II (COL2A1). This study demonstrates, for the first time, that PN exerted anti-inflammatory effects by partially inhibiting the NF-κB pathway and increasing the Smad2/3 pathway. Based on our findings, PN can potentially serve as a treatment for OA.

## 1. Introduction

Osteoarthritis (OA), for which there is no known cure, is a degenerative joint disease that significantly impacts people’s lives and the economy. Age, obesity, knee malalignment, biomechanical stress on joints, low-grade systemic inflammation, etc., all contribute to the onset of OA [1]. From 1990 to 2019, the global average number of years a person is disabled due to OA increased by 114%, placing enormous financial strain on societies worldwide [2]. OA joints exhibit several pathological abnormalities, including loss of cartilage in the articular region, subchondral bone enlargement, osteophyte formation, varying degrees of synovial inflammation, and ligament degeneration. Although cartilage deterioration [3] is a hallmark of OA, inflammation can contribute to its development by promoting cartilage deterioration and reducing the regenerative capacity of cartilage [4].

Interleukin−1β (IL−1β), a pro-inflammatory cytokine, has been implicated in the degradation of articular cartilage through the induction of chondrocyte hypertrophy [5]. Chondrocyte hypertrophy involves significant changes in gene expression patterns within chondrocytes. These changes are accompanied by the production of hypertrophic markers, including matrix metalloproteinases (MMPs), and alterations in the remodeling of the extracellular matrix (ECM). In OA, dysregulated chondrocyte hypertrophy can contribute to pathological remodeling of the ECM, cartilage degradation, and joint dysfunction [6]. While several immune factors, including TNF-α, IL−6, and IL−17, have been associated with the pathogenesis of OA, their significance is relatively less compared to IL−1β. Extensive research has highlighted IL−1β as a key player in the development and progression of OA due to its potent pro-inflammatory effects and ability to induce catabolic processes in chondrocytes. Clearly, IL−1β stands out as a crucial immune factor in the context of OA pathogenesis [7].

In OA patients, elevated levels of IL−1β were detected in the synovial fluid, synovial membrane, cartilage, and subchondral bone [8]. In addition, elevated expression of IL−1β receptor type 1 has been observed in chondrocytes in OA patients compared to healthy individuals, leading to enhanced OA progression through the upregulation of catabolic factors, such as nitric oxide (NO) and cyclooxygenase-2 (COX-2) [9,10,11,12]. These enzymes stimulate synoviocytes, which further promote the production of catabolic factors, resulting in erosion, cracking, and fibrillation of the articular cartilage [13]. The repeated cycles of inflammation and catabolism disrupt cartilage homeostasis, leading to irreversible matrix breakdown and the development of OA [14]. The nuclear transcription factor κB (NF-κB) pathway plays a crucial role in mediating the catabolic effects of IL−1β by regulating genes associated with inflammation, the immune response, and apoptosis. Consequently, targeting the NF-κB pathway holds therapeutic potential for managing OA [15]. Additionally, IL−1β has been extensively studied for its ability to inhibit matrix synthesis, promote cartilage catabolism, and influence transforming growth factor-β (TGF-β) signaling [16]. Proinflammatory conditions hinder the protective TGF-β/Smad2/3 pathway, which otherwise inhibits chondrocyte hypertrophy and MMP13 production [17,18]. Researchers are currently focusing on developing disease-modifying OA medications (DMOADs) that can restore TGF-β signaling and preserve ECM synthesis [19]. However, the clinical efficacy and potential adverse effects of these treatments are still under investigation, necessitating the urgent discovery of new OA therapies.

Currently, the most prevalent treatments for OA are surgical and nonpharmaceutical procedures. However, the lifespan of prostheses is still fairly limited, and the only effective treatments for OA are lifestyle changes and joint replacement surgery [20]. If surgery is to be avoided, viscosupplementation is a viable alternative if the more conservative treatments fail [21]. Viscosupplementation is classified as localized therapy, specifically intra-articular (IA) pharmaceutical therapy. Viscosupplementation is a valid technique for alleviating OA-related pain and improving joint function. While some clinical trials claimed that hyaluronic acid (HA) had a disease-modifying effect, subsequent meta-analyses cast doubt on this assertion [22]. In addition to protecting the cartilage surface from further damage, an effective IA viscosupplement for OA would return the chondrocytes to homeostasis by creating a more physiological microenvironment and resupplying them with nutrition. From this perspective, OA has been treated with an upcoming technology called Polynucleotides Highly Purified Technology (PN-HPT), which is a promising technology that has previously shown potential to improve skin rejuvenation, revitalization, and tonification of the face and body [23]. A viscosupplement produced by this technology is polynucleotides consisting of DNA fragments. IA-PN involves the administration of a fixed formulation of intra-articular polynucleotides. IA-PN has been shown to reduce pain and improve function in patients with knee OA [24]. PN is also known to exhibit a chondroprotective effect, reducing cartilage degradation. Cesare et al., demonstrated that patients with knee OA experience improved knee function and decreased pain after a brief cycle of IA therapy that includes PN in fixed combination with high molecular weight HA [25]. PN provides chondrocytes and mesenchymal cells with nitrogenous bases as well as nucleoside and nucleotide progenitors [26]. Although PN has its origins in regenerative medicine, no studies on its ability to regenerate chondrocytes in OA have been published. However, according to Hwang et al., in a rat model with chronic infraspinatus tears, histological and biomechanical tests revealed that polydeoxyribonucleotide (PDRN) and PN administered to the subacromial region improved human rotator cuff recovery, which remains subpar, even after a successful cuff replacement.

Furthermore, the effects of PN can last even longer than those of PDRN because PN creates a favorable environment for injured areas by synthesizing ECM [27]. Polynucleotides have the potential to slow cartilage deterioration in OA patients by decreasing the production of catabolic mediators and increasing the levels of anabolic markers [28]. Although numerous animal and clinical studies [24,29] have demonstrated the efficacy of PN in promoting cartilage regeneration and suppressing OA inflammation and pain, the mechanism of action remains unknown. Consequently, this study aimed to investigate the molecular mechanism underlying the enhanced anti-inflammatory response and cartilage repair. The PN used in this study was polynucleotide sodium (CONJURAN) purchased from PharmaResearch Co. Ltd. (Gyeonggi-do, South Korea), wherein the main component of it is a safe DNA material extracted from reproductive cells of salmon and refined with a high purification process, such as PN-HPT, to make it suitable for injection into a human body part, such as joints. The expert consensus illustrates the value of natural-origin, highly purified polynucleotides (PN-HPT) as a bio-stimulatory booster [30]. In this study, we aimed to investigate the positive effects and underlying mechanisms of the IA viscosupplement PN using an in vitro 2D cell model. Considering the absence of a standardized OA cell model that comprehensively captures the various aspects of the disease, we recognized the widely accepted use of chondrocytes isolated from OA patients for studying cartilage inflammation and catabolism. However, our objective was to address this limitation by establishing a novel model specifically for evaluating the efficacy of the viscosupplement. To accomplish this, we utilized commercially available human articular chondrocytes (HC-a) and stimulated them with IL−1β to simulate the inflammatory environment associated with OA. By focusing on the NF-κB/TGF-β/Smad family member 2/3 (Smad2/3) pathway, we aimed to elucidate the mechanism of action of PN.

## 2. Results

### 2.1. IL−1β Induces the Expression of Proinflammatory Markers in HC-a in a Concentration and Time-Dependent Manner

IL−1β is one of the most important proinflammatory cytokines involved in the pathophysiology of OA. Consequently, we evaluated the viability of HC-a treated with varying concentrations of IL−1β and discovered that IL−1β decreased cell viability in a concentration and time-dependent manner (Figure 1A,B). For subsequent experiments, a concentration of 10 ng/mL was utilized because previous research demonstrated that this concentration not only caused a significant decrease in HC-a viability but was also the most effective in activating MMP13 and inhibiting COL2A1 in HC-a (Figure 1C). Since the dose was determined, HC-a cells were exposed to IL−1β at 10 ng/mL for varying amounts of time. In response to IL−1β stimulation, a time-dependent increase in MMP13 and COL2A1 expression levels was detected (Figure 1F). After 24 h and 48 h of stimulation, the expressions of MMP13 and COL2A1 were significantly altered compared to other periods. In subsequent experiments, we examined the effects of IL−1β stimulation of HC-a at 10 ng/mL for 24 h.

### 2.2. Effect of PN on HC-a Viability

The cytotoxic effects of PN on HC-a were determined at various concentrations (1, 10, 100, and 1000 μg/mL) for 24 h using the cell counting kit-8 (CCK8) assay. The viability of HC-a cells was significantly enhanced at doses of PN ranging from 1 to 100 µg/mL. Therefore, based on previous research, for subsequent experiments, concentrations of 1, 10, and 100 μg/mL PN were used and HC-a chondrocytes were exposed to PN 24 h after IL−1β stimulation [31], demonstrating the administration of PN to patients with OA grades I to III, and thus replicating the inflammatory environment observed in these patients. Overall, the results showed that PN exhibited a dose-dependent inhibition of IL−1β-induced cytotoxicity at concentrations of 1, 10, and 100 µg/mL (Figure 2B).

### 2.3. MMP13 and Nitric Oxide (NO) Assay

In order to investigate the protective mechanism of PN in regulating IL−1β-activated inflammatory mediators in HC-a, the levels of MMP13, a matrix-degrading protein associated with OA, were measured in the presence of IL−1β alone and IL−1β combined with different concentrations of PN. The results showed that IL−1β treatment significantly increased MMP13 activity compared to controls. However, the addition of PN at 1, 10, or 100 µg/mL significantly decreased MMP13 activity, with the highest concentration of PN reducing MMP13 activity by over 70% (Figure 3A). In addition, the study evaluated the effects of PN on IL−1β-induced production of nitric oxide (NO), a known mediator of inflammation in OA. HC-a treated with IL−1β exhibited a significant increase in NO levels compared to the control group. However, when HC-a cells were treated with 100 µg/mL PN in combination with IL−1β, NO production was significantly inhibited in a dose-dependent manner (Figure 3B). Overall, these findings suggest that PN has a protective effect in regulating IL−1β-induced inflammatory mediators, such as MMP13 and NO, in HC-a.

### 2.4. Effects of PN on Inflammation and Hypertrophy in IL−1β-Induced HC-a

IL−1β stimulation revealed that PN could reduce MMP13 and nitric oxide levels via ELISA and Griess reagent assays; therefore, we investigated whether PN could also reduce the gene and protein levels of MMP13, iNOS, and other hypertrophic and inflammatory mediators released in response to IL−1β stimulation. As a result, the gene expression of matrix-degrading enzymes, such as MMP13 and MMP3, and inflammatory mediators, such as iNOS and COX-2, were evaluated. Real-time PCR was utilized to assess gene expression. Treatment with IL−1β significantly increased the expression of all genes, while treatment with PN significantly inhibited the expression of MMP13, MMP3, iNOS, and COX-2 (Figure 4A). There was a dose-dependent inhibitory effect, with the greatest inhibition occurring at 100 µg/mL. Gene expression was unaffected by 1 µg/mL PN treatment. However, MMP13, MMP3, and iNOS gene expression were significantly reduced at a PN concentration of 10 µg/mL. All inhibitory effects exhibited a dose-dependent manner. Since we discovered that the expression of the aforementioned genes was suppressed in response to PN treatment, we then examined protein levels using Western blotting. After IL−1β stimulation, we observed an increase in the levels of all four proteins, which corresponded to what was observed at the mRNA level (Figure 4F). Treatment with 100 µg/mL PN significantly decreased protein concentrations. Similar to the reduction in gene expression, protein levels of MMP13, MMP3, iNOS, and COX-2 decreased dose-dependently after PN treatment at 100 µg/mL. Additionally, 10 µg/mL showed marginal significance for MMP3, iNOS, and COX-2, whereas the protein levels did not show any significance when treated with 1 µg/mL PN. Immunofluorescence observations for iNOS were in agreement with Western blotting results (Figure 4K). In conclusion, 100 µg/mL of PN controls IL−1β-induced MMP13, MMP3, iNOS, and COX-2 production in HC-a.

### 2.5. PN Relieves IL−1β-Induced Inflammation of HC-a through the NF-κB Signaling Pathway

We investigated whether the protective effects of PN on HC-a are related to its modulation of the NF-κB signaling pathway in cells treated with IL−1β. We examined the phosphorylated and total protein levels of p65 and IκBα in whole-cell lysates using Western blotting and 100 µg/mL of PN, which has been shown to have the greatest effect in decreasing hypertrophic and inflammatory mediators. IL−1β treatment resulted in a significant increase in pp65 protein expression relative to the control group, indicating activation of the pp65 signaling pathway. However, PN treatment effectively suppressed pp65 expression, indicating that it inhibits the NF-κB signaling pathway (Figure 5A). In order to gain a deeper understanding of PN’s mechanism of action in IL−1β-mediated inflammation in HC-a, we employed 5HPP-33, an NF-κB signaling inhibitor. Our findings revealed that 5HPP-33 partially inhibited the activation of the NF-κB pathway and effectively prevented the IL−1β-induced expression of pp65. Notably, we also verified the impact of PN on pp65 expression in the presence of the inhibitor, further substantiating its effect (Figure 5F). Immunofluorescence staining of pp65 in the presence and absence of the inhibitor additionally confirms the suppression of the NF-κB pathway observed by Western blotting (Figure 5H). Furthermore, treatment with the NF-κB signaling inhibitor increased the effect of PN in inhibiting MMP13, MMP3, COX-2, and iNOS expression, thus establishing that PN suppresses the protein levels of MMP13, MMP3, COX-2, and iNOS (Figure 5I) via the NF-κB pathway. PN significantly inhibits the NF-κB pathway, which is responsible for the release of inflammatory mediators. By inhibiting NF-κB pathway activation in HC-a, PN subsequently reduced the production of inflammatory mediators and cytokines.

### 2.6. Effects of PN on ECM Synthesis in IL1β-Induced HC-a

Since we had previously demonstrated that PN can inhibit MMP3 and MMP13, we sought to determine whether PN could also increase the production of TGF-β, ACAN, and COL2A1 production. mRNA (Figure 6A) and protein levels (Figure 6E) of TGF-β, COL2A1, and ACAN were significantly higher in cells treated with PN than in control IL−1β -treated HC-a in this study. All changes in IL−1β expression were completely reversed by PN treatment, particularly at a dose of 100 µg/mL. The observed improvements in TGF-β, ACAN, and COL2A1 suggest that PN plays a significant role in the regulation of ECM production and chondrocyte growth based on these findings. According to immunofluorescence, COL2A1-positive proteins were primarily located in the cytoplasm, supporting the results of the Western blot analysis (Figure 6I).

### 2.7. PN Increases the Phosphorylation of Smad2/3 in HC-a

We also investigated whether or whether not the chondrocyte Smad protein is modified by inflammation caused by IL−1β. IL−1β has been observed to disrupt Smad signaling in chondrocytes. Therefore, we investigated whether it was possible to use PN to reestablish Smad signaling in HC-a (Figure 7A) using a potent, selective inhibitor of TGF-β, namely RI kinase inhibitor LY-364947 (Calbiochem, Sigma Aldrich), of the Smad pathway. HC-a treated with PN exhibited an increase in phosphorylation of Smad2/3, which was inhibited by pretreatment with the inhibitor LY-364947. When LY-364947 was also administered, the protective effects of PN were reduced, again demonstrating that PN activity occurs along the Smad pathway. We analyzed whether the PN increased TGF-β, COL2A1, and ACAN through the Smad2/3 pathway (Figure 7D). Immunofluorescence staining of p-Smad2/3 in the presence and absence of the inhibitor is consistent with the suppression of the Smad pathway seen via Western blot (Figure 7C). The findings revealed that IL−1β-treated HC-a benefited from PN because it stimulated ECM production. Our findings indicate that in HC-a treated with IL−1β, PN is essential for restoring Smad signaling and favoring the production of chondrogenic markers.

## 3. Discussion

As the population ages and more people struggle with obesity, OA has become a more severe health problem. Symptomatic OA affects approximately 240 million people worldwide, including 10% of men and 18% of women aged 60 and older [32]. Although the social, economic, and physical consequences of OA are severe, there is currently no treatment other than total joint replacement for advanced cases [33]. Nonsteroidal anti-inflammatory drugs (NSAIDs) are frequently used as first-line treatments due to their exceptional pain-relieving properties. NSAIDs cannot slow the progression of the disease, and patients may experience undesirable side effects, such as dyspepsia, gastrointestinal risk, and thrombus formation [34]. Therefore, there is an urgent need for agent therapies that can significantly improve the etiology of OA while maintaining a high degree of safety. Multiple systematic studies have shown that viscosupplementation reduces pain and inflammation [35]. Polynucleotide (PN) as a viscosupplement for the treatment of OA is a recent development [36]. Since the 1990s, PN has been applied to the skin to promote healing [37]. However, since the 2010s, it has been standard therapy for OA. IA PN is used to nourish cartilage and the surrounding joint environment, restoring a healthy balance to chondrocytes [38]. PN is a polymer that forms a three-dimensional gel when combined with a substantial amount of water. These characteristics make it ideal for IA administration, which has been shown to dramatically alleviate OA pain and inflammation [29]. It is well established that PN is a physiologically significant component of the extracellular microenvironment. The constituents of the cellular matrix are nitrogen bases, nucleotides, and nucleosides. The provision of these substrates to chondrocytes promotes the physiological healing of cartilage as an additional benefit [39]. Being a naturally occurring pure form of DNA polymer in humans, PN offers minimal adverse effects, such as foreign body response, and has been demonstrated to be non-toxic and protective for chondrocytes in vivo [40,41]. However, the precise mechanisms by which PN exerts its protective effects against OA remain unclear. Therefore, this study aimed to evaluate the efficacy and mechanism of PN as a viscosupplement in an in vitro IL−1β-induced cell-based model. Although PN is highly biocompatible and has a low risk of adverse effects, as shown in Figure 2A, a slight decrease in cell viability was observed at high concentrations (1000 µg/mL). This decrease may be attributed to increased viscosity, hindered nutrient passage, and exerted external stress on the cytoskeleton, ultimately affecting cell viability. These findings highlight the importance of investigating the optimal concentration of PN for effective cartilage healing and further elucidating its mechanisms of action.

Previous research has shown that an increase in the production of inflammatory cytokines [7] is a significant factor in the development of OA. Among these cytokines, IL−1β has the greatest effect on the disease. MMP synthesis increases in the presence of IL−1β, followed by ECM degradation [42]. IL−1β at 10 ng/mL significantly increases inflammation and catabolic processes while simultaneously suppressing chondrocyte anabolism [43], as demonstrated by our findings. IL−1β increases the expression of proinflammatory mediators, such as MMP3, MMP13, iNOS, and COX-2. Typically, deterioration of articular cartilage is primarily caused by members of the MMP family, particularly MMP3 and MMP13. By breaking down collagen and altering the proteoglycan structure in the cartilage matrix, these proteins contribute to the onset of OA [44]. NO in chondrocytes is mainly produced by iNOS, and NO overproduction has been linked to the death of chondrocytes and synoviocytes, making it a key proinflammatory factor in the development of OA [45]. The production of MMPs and other inflammatory cytokines is stimulated by NO. In this study, it has been demonstrated that PN inhibits the inflammatory response of IL−1β stimulated HC-a by decreasing the expression of pro-inflammatory cytokines, such as MMP3, MMP13, iNOS, COX-2, and NO production.

Our research demonstrated that PN inhibited the expression of MMP3 and MMP13. IL−1β stimulation is also believed to be characterized by chondrocyte modifications that resemble hypertrophy [46]. The NF-κB expression factor is a crucial regulator of the IL−1β-induced inflammatory response. NF-κB (p65) is a cytoplasmic protein NF-κB that interacts with its inhibitor κB (IκB). Inhibitory kappa B kinases can cause the phosphorylation, ubiquitination, and proteolytic destruction of IκBα in response to stimulation from cytokines like IL−1β [47]. Simultaneously, p65 travels from the cytoplasm to the nucleus, where it attaches to the target gene and stimulates transcription [48]. Activation of the NF-κB pathway results in the production of iNOS and COX-2, two genes involved in the inflammatory response [49]. These inflammatory mediators increase the production of MMP3 and MMP13; these enzymes degrade ECM and inhibit its formation [50,51]. Furthermore, it was observed that PN treatment inhibited p65 phosphorylation, suggesting its potential role in modulating the NF-κB signaling pathway. In addition, to comprehensively assess the molecular mechanisms underlying this modulation, future investigations will focus on determining the binding specificity of PN towards various NF-κB elements. Inhibition of the IKK activity by 5HPP-33, a known NF-κB pathway inhibitor, has been extensively investigated, demonstrating its ability to block NF-κB DNA binding and transcriptional activity, thus exerting anti-inflammatory effects [52,53]. In addition, 5HPP-33, a thalidomide derivative, has been reported to inhibit the DNA-binding activity of transcription factors like Sp1, contributing to its inhibitory effects on NF-κB-mediated gene expression [54]. 5HPP-33’s capacity to target NF-κB signaling highlights its potential for regulating inflammatory responses [55]. In this study, the NF-κB pathway inhibitor 5HPP-33 was utilized, and its combination with PN demonstrated a significant enhancement of PN’s anti-inflammatory activity as well as a suppressive effect on the production of iNOS, COX-2, MMP3, and MMP13 proteins. These findings suggest that PN’s protective effects may, in part, be mediated by the inhibition of the NF-κB signaling pathway, as depicted in Figure 5. Notably, the anti-inflammatory properties of PN resemble those of PDRN, a registered drug derived from PN. PDRN engages adenosine A2A receptors, suppressing inflammatory responses and pro-inflammatory cytokine production [56]. It generates nucleotides and nucleosides, which support DNA synthesis and facilitate normal cell proliferation and growth. Similar mechanisms may underlie the observed effects of PN. Overexpression of catabolic factors can initiate an inflammatory reaction, and the positive feedback loop is fueled by proinflammatory mediators that aid in ECM degradation [57]. Articular cartilage relies on COL2A1 and ACAN, two primary components of the ECM, to maintain its pliability and rigidity [58]. The primary symptom of early arthritis is believed to be a decline in ACAN [59] According to our findings, PN prevented the breakdown of the ECM by stopping the deterioration of COL2A1 and ACAN. According to the underpinning research, PN treatment greatly increased the transcriptional and translational expression of TGF-β, COL2A1, and ACAN in IL−1β induced HC-a. Chondrocytes make COL2A1 and release it into the ECM as a defense mechanism. Therefore, the capacity of PN to support cartilage repair may be linked to the upregulation of COL2A1 expression. Enhanced TGF-β expression in HC-a is associated with improved ECM production, differentiation, adhesion, and movement [60]. Furthermore, in chondrocytes, Smad2 and Smad3 function as intracellular effectors of the TGF-β signaling pathway. Smad2 regulates gene expression indirectly, while Smad3 binds DNA directly, playing an important role in modulating TGF-β signaling. Chondrocyte proliferation and differentiation can be controlled by Smad2 and Smad3, with Smad3 suppressing terminal hypertrophic differentiation. To counteract the anti-inflammatory effects of TGFβ in chondrocytes, inflammation dampens the Smad2/3 pathway [61,62]. The gene expression of COL2A1 and ACAN is promoted by PN via TGFβ1/SMAD signaling (as shown in Figure 8). The activation of the TGFβ-SMAD2/3 pathway has been demonstrated to occur through a membrane signaling complex consisting of the type I TGFβ receptor and the type II collagen receptor, with COL2A1 as the activator [63]. Our data indicated that PN protected HC-a by substantially upregulating TGF-β and phosphorylated Smad2/3 and cartilage-specific genes (Figure 6 and Figure 7). The TGF-β receptor inhibitor LY-364947 was used to confirm the importance of the TGF-β/Smad2/3 signal in the protective effects of PN in HC-a. It has also been shown that LY-364947 exerts its protective effects on HC-a by modulating the signal from TGF-β/Smad2/3 and that this impact cannot be reversed by PN.

In the treatment of OA, PN has been proven to be an effective alternative to HA. PN has exceptional anti-inflammatory actions that promote cartilage recovery [46]. PN filler injections were used to heal cartilage in a rat model of OA with positive results. Additionally, patients suffering from knee OA experienced relief from their pain more quickly and effectively with PN than with traditional or cross-linked HA [28]. Highly purified polynucleotides (PN-HPTTM) of natural origin have been shown to have lasting benefits in a 2-year, double-blind study [64]. Also, in a rabbit model with persistent rotator cuff tear, Dong and Yong found that a combination of PDRN and microcurrent decreased tendon tear size and improved collagen fiber regeneration, cell proliferation, angiogenesis, gait speed, and overall walking distance [65]. Moreover, the impacts of PN can last even longer than PDRNs because this biomaterial offers an appropriate microenvironment for injured areas through the formation of the ECM. Ahreum Baek et al. demonstrated in a previous study that, in an OA cell model using a chondrosarcoma cell line (SW1353), PN had a greater anti-inflammatory effect than PDRN. Our results support their findings [44]. According to their findings, PN treatment had a greater anti-inflammatory effect than PDRN treatment. The authors did not examine any route or discuss any mechanism associated with the observed decrease in inflammatory cytokines and chemokines. Our findings provide the first comprehensive evidence that PN suppresses the inflammatory response and ECM breakdown in HC-as via the NF-κB and Smad pathways (as shown in Figure 8).

The primary motivation for employing a 2D in vitro cell-based model in this study was to evaluate the mechanism of a viscosupplement (PN) for which clinical studies have been completed; however, the mechanism of action has not been adequately discussed. In addition, very little is known about the mechanism by which PN exerts these properties, despite the fact that it is said to have high regenerative properties and an efficient repair mechanism. Since the 1990s, PN has been utilized in skin regeneration therapies and OA treatment. When applied for skin regeneration, PN has been shown to stimulate the production of a stronger ECM, as well as collagen, and elastin, which restores youthful midface convexity in a consistent and durable manner. In addition, since PNs are efficient water binders that reorient their spatial structure and orientation after binding water molecules in dermal tissues, they are also effective water binders. The resultant three-dimensional viscoelastic PN gel demonstrated prolonged hydration of the ECM. Based on clinical studies demonstrating the role of PN in rotator cuff healing, skin rejuvenation, and management of OA, we investigated two main signaling pathways. The involvement of PN in other signaling pathways is unknown and necessitates further investigation. When evaluating IA viscosupplements, it is critical to understand the advantages and disadvantages of the IA in in vitro models. Therefore, 2D models of cells, such as synoviocytes or chondrocytes, that respond to cytokine stimulation (typically IL−1β) are the optimal choice for screening anti-inflammatory or chondroprotective molecules for the evaluation of IA delivery systems [66]. In vitro, 2D models of OA allow low-cost, high-throughput analysis and exact control over specific factors [67]. However, they are linear in design and do not accurately depict physiological conditions. It is also true that in vitro tends to dedifferentiate into fibroblasts [68]. We observed that the characteristics of chondrocytes and the expression of chondrogenic genes, such as COL2A1 and ACAN, remained essentially unchanged over three generations [69]. In subsequent studies, chondrocytes were only used if they had been cultured for no more than three passages.

## 4. Materials and Methods

### 4.1. Reagents

Polynucleotide (PN) from this study (CONJURAN, PharmaResearch Co., Ltd., Gyeonggi-do, South Korea) was supplied in prefilled sterile syringes containing 2 mL of a clear, colorless solution. At 37 °C, PN was dissolved directly in a standard cell culture medium. When cells were in a standard cell culture medium, the optimal concentrations of this solution were determined. Cell culture media 1X Dulbecco’s Modified Eagle’s Medium (DMEM), Penicillin streptomycin (pen strep), and fetal bovine serum (FBS) were supplied from Gibco (Thermo Fisher Scientific, Waltham, MA, USA). R&D systems (Minneapolis, MN, USA) provided human recombinant IL−1β, which was dissolved in PBS containing 0.5% bovine serum albumin. Antibodies against type II collagen (COL2A1) (Abcam, Boston, MA, USA), aggrecan (ACAN) (Abcam, USA), COX-2 (Abcam, Boston, MA, USA), iNOS (Santa Cruz, TX, USA), MMP3 (BioLegend, San Diego, CA, USA), MMP13 (Bioss, Woburn, MA, USA), Total-Smad (Bioss, Woburn, MA, USA), p-Smad2/3 (Invitrogen, CA, USA), p65 (cell signaling, Danvers, MA, USA), pp65 (cell signaling, Danvers, MA, USA), IκBα (cell signaling, Danvers, MA, USA), and p-IκBα (cell signaling, Danvers, MA, USA) were purchased. Additionally, we purchased conjugated goat anti-rabbit secondary antibody (H+L) (Novex life technologies, Thermo Fisher Scientific, Waltham, MA, USA) and goat antimouse IgG (ZyMax, Thermo Fisher Scientific, Waltham, MA, USA).

### 4.2. Cell Culture

The human articular chondrocyte (HC-a) cell line was purchased from ScienCell (Carlsbad, CA, USA) (#4650) [70,71]. Cells were grown in 1X DMEM supplemented with 1% pen strep and 10% heat-inactivated FBS. When the cell density reached 80%, they were transferred to a culture plate and passed. In our studies, we used chondrocytes between passages 2 and 3. Please note that HC-a donor specifications are not available. For further information on the characterization of HC-a cells, we recommend contacting the manufacturer, ScienCell.

### 4.3. Chondrocyte Induction In Vitro

When cells reached 80% confluence, we adopted a pretreatment approach by exposing HC-a to IL−1β for 24 h, followed by the administration of PN for the subsequent 24 h. The IL−1β concentration was 10 ng/mL, while the concentrations of PN used were 1, 10, and 100 µg/mL. We chose to examine post-treatment of PN after exposure to proinflammatory cytokines rather than pretreatment before stimulation with inflammatory cytokines because pretreatment may not reflect clinical reality. Due to the prolonged exposure to an inflammatory environment, patients with Kellgren–Lawrence grade II or grade III knee OA frequently experience pain and limited mobility. Therefore, we investigated whether administering PN after exposure could aid in reducing inflammation in HC-a.

### 4.4. Measurement of Cell Viability

CCK-8 test kit (Dojindo, Japan) was used to determine cell viability after IL−1β and PN treatment. In summary, cells were seeded in 96-well plates at a density of 1 × 10^4^ cells/well, stimulated with IL−1β for 24 h, and then treated with or without PN at concentrations of 1, 10, and 100 µg/mL for 24 h. After appropriate treatment, according to the experimental grouping, 10 µL of CCK-8 solution was added to each well and incubated for 2 h at room temperature. The level of the dye formed was measured with a BioTek Synergy HTX multimode plate reader (Agilent Technologies, Santa Clara, CA, USA) with the absorbance at 450 nm in each well.

### 4.5. Enzyme-Linked Immunosorbent Assay (ELISA)

Next, using the biorbyt MMP13 ELISA Kit, we analyzed the ability of PN to inhibit MMP13 activity in vitro (St. Louis, MO, USA). Cells were plated in 12-well plates and treated with IL−1β (10 ng/mL) alone or with PN (1, 10, and 100 µg/mL) at various concentrations for 24 h. Following the manufacturer’s instructions, MMP13 activity was determined using the biorbyt MMP13 Assay Kit (St. Louis, MO, USA) on the culture medium.

### 4.6. Nitric Oxide (NO) Assay

The nitric oxide colorimetric assay was conducted following a previously established protocol [72]. To assess nitric oxide (NO) production, the Griess reagent was employed to quantify the level of nitrite, a stable metabolite of NO, in the culture medium, serving as an indicator of NO production. The levels of nitrite were determined by comparing them to a standard curve generated using sodium nitrite dissolved in distilled water. Following 24 h of stimulation with IL−1β (10 ng/mL), the cells were treated with PN for an additional 24 h. Equal volumes of the supernatant and Griess solution were combined in a 96-well plate and incubated for 30 min. The samples were then analyzed using a spectrophotometer set to 540 nm.

### 4.7. PCR

To assess the effects of IL−1β and PN on the transcription of MMP3, MMP13, ACAN, COL2A1, TGF-β, COX-2, iNOS, and the housekeeping enzyme glyceraldehyde-3-phosphate dehydrogenase (GAPDH) (primer details in Table 1), HC-a cells grown to 80% confluence on plates with/without PN were homogenized with RNAiso Plus reagent (TaKaRa Bio Inc., Kusatsu, Japan). The total RNA was extracted, and 0.5 μg RNA aliquots were reverse-transcribed in 20 μL buffer containing 5 × AMV reverse transcriptase; 2.5 μM poly dT; 1 mM each of dATP, dCTP, dGTP, and dTTP; 20 U of RNase inhibitor; and 20 U of AMV RT. Reverse transcription was performed using the following conditions: initial incubation at room temperature for 10 min and then 42 °C for 15 min, 97 °C for 5 min, and 5 °C for 5 min in a GeneAmp PCR System 2700 (Applied Biosystems, Foster City, CA, USA). Next, aliquots of cDNA were amplified in AccuPower^®^ GreenStar PCR premix (Bioneer Co., Daejeon, South Korea) using the MyGenieTM 96/384 thermal cycler system (Bioneer Co., Daejeon, South Korea).

### 4.8. Whole-Cell Lysate Preparation and Western Blot Analysis

At the end of the indicated treatment, cells were lysed with the radioimmunoprecipitation assay (RIPA) and lysis solution containing protease and phosphatase inhibitor cocktails that had been cooled. The samples were incubated on ice for 30 min before being centrifuged at 12,000 rpm and 4 °C for 10 min. The protein concentration of the whole cell lysate was determined using a bicinchoninic acid (BCA) protein assay kit (Thermo Scientific, Waltham, MA, USA) per the manufacturer’s instructions. The protein was then combined with a sodium dodecyl sulfate sampling buffer, followed by a 5-min incubation at 95 °C. A total of 20 μg protein was loaded onto sodium dodecyl sulfate-polyacrylamide gels (SDS-PAGE) and separated by electrophoresis for each sample. The proteins were then transferred to PVDF membranes (Bio-Rad, Hercules, CA, USA); the membranes were blocked with 5% skim milk (DifcoTM, Becton Drive Franklin Lakes, NJ, USA) for 2 h and overnight incubated with primary antibodies directed for iNOS (1:1000), COX-2 (1:1000), MMP13 (1:1000), MMP3 (1:1000), ACAN (1:1000), COL2A1 (1:1000), p65 (1:1000), pp65 (1:1000), IκBα (1:1000), TGF-β (1:1000), total Smad2/3 (1:1000), p-Smad2/3 (1:1000), and β-actin (1:5000) at 4 °C. The membranes were incubated for 2 h at room temperature with the appropriate enzyme-linked secondary antibodies. Blots were visualized using enhanced chemiluminescence (ECL) solution as per the manufacturer’s instructions. The band intensity was quantified using ImageJ software (National Institutes of Health, Bethesda, MD, USA).

### 4.9. Inhibitor Treatment

ACs (1 × 10^6^/mL) were allowed to grow for 2 to 3 days in 10% DMEM and FBS supplemented with penicillin (100 U/mL) before being used in the experiments. HC-a were serum starved for 6 h to equilibrate all cells, pretreated with 10 μM of the NF-kB signaling inhibitor 5HPP-33 (Refer Appendix A for details on 5HPP-33 concentration determination) (Calbiochem, La Jolla, CA, USA) or 10 μM TGF-β RI kinase inhibitor LY-364947 (Calbiochem, La Jolla, CA, USA) for 2 h, and then exposed to IL−1β (10 ng/mL) in the presence or absence of PN (100 μg/mL) for 24 h.

### 4.10. Immunofluorescence

HC-a cells grown on coverslips were fixed with 4% paraformaldehyde in PBS for 15 min, permeabilized with 0.1% Triton X-100 for 15 min, and then blocked with 5% BSA in PBS for 30 min. The coverslips were then incubated for 1 h at room temperature with primary antibodies against pp65 and p-Smad2/3 at a 1:100 dilution and secondary antibodies at a dilution of 1:500. The cells were then washed with PBS and mounted in glycerol at a concentration of 70%. Fluorescence micrographs were obtained with a BioTek Lionheart FX Agilent microscope (Seoul, Korea). Lastly, nuclei were stained with DAPI solution.

### 4.11. Statistical Analysis

All experiments in this study were conducted in triplicate to ensure reproducibility. The data are presented as the mean ± standard error. GraphPad Prism 9.0 software (GraphPad Software Inc., La Jolla, CA, USA) was utilized for generating the graphical representations of the data. Statistical analysis was performed to determine the significance of differences between two or more groups. The student’s *t*-test was employed for comparisons between two groups, while, for multivariable analyses involving multiple groups, analysis of variance (ANOVA) with Tukey’s multiple comparison test was implemented. Statistical significance was defined as *p* < 0.05 for one asterisk (#, *, @), *p* < 0.01 for two asterisks (##, **, @@), *p* < 0.001 for three asterisks (###, ***, @@@), and *p* < 0.0001 for four asterisks (####, ****, @@@@).

## 5. Conclusions

In conclusion, we found that PN decreased the expression of MMP3, MMP13, iNOS, and COX-2 in HC-a while increasing TGF-β, COL2A1, and ACAN expression. In this preliminary study, we demonstrate that PN protects chondrocytes by inhibiting key molecular events involved in inflammation through NF-κB activation and matrix synthesis through the Smad pathway. In addition, our research reveals novel information regarding the potential of PN as an anti-inflammatory and chondroprotective agent.

## Figures and Tables

**Figure 1 ijms-24-12282-f001:**
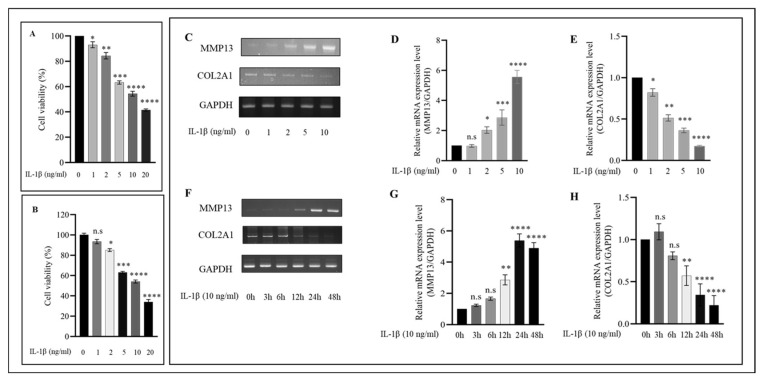
HC-a cell viability was examined in response to various concentrations of IL−1β after stimulation for (**A**) 24 and (**B**) 48 h. Additionally, (**C**) IL−1β concentration-dependent mRNA expressions of MMP13 and COL2A1 were evaluated using RT-PCR, and (**D**,**E**) quantitative analysis was performed. Following identification of the optimal IL−1β concentration (10 ng/mL), (**F**) the expression levels and (**G**,**H**) quantitative analyses of MMP13 and COL2A1 were evaluated at different time points (0, 3, 6, 9, 12, 24, and 48 h) using RT-PCR. The data, presented as the mean ± standard deviation (n = 3), were analyzed for statistical significance using the following significance notations: n.s (no significance), * *p* < 0.05, ** *p* < 0.01, *** *p* < 0.001, and **** *p* < 0.0001 compared to the control group (unstimulated cells).

**Figure 2 ijms-24-12282-f002:**
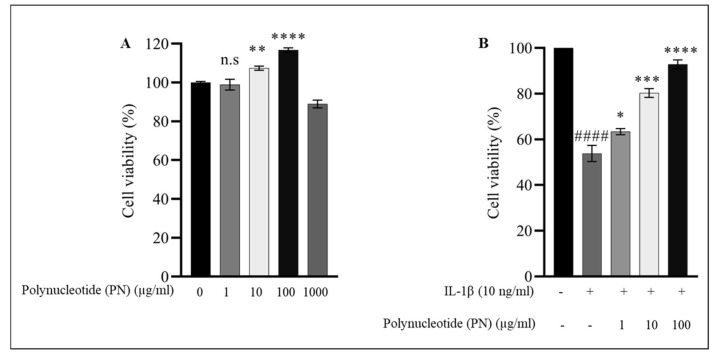
(**A**) Cell viability of HC-a chondrocytes in response to varying concentrations of polynucleotide (PN). The viability of HC-a chondrocytes was measured after exposure to different concentrations of PN. (**B**) Cell viability of IL−1β-treated HC-a chondrocytes in response to PN. The cell viability of HC-a chondrocytes, pre-treated with IL−1β, was assessed following exposure to PN. Data are presented as mean ± standard deviation (n = 3). Statistical analysis revealed significant differences, with the following symbols denoting significance values: n.s (no significance); #### *p* < 0.0001 compared to the control group (unstimulated cells); * *p* < 0.05, ** *p* < 0.01, *** *p* < 0.001, and **** *p* < 0.0001 compared to the IL−1β-treated group.

**Figure 3 ijms-24-12282-f003:**
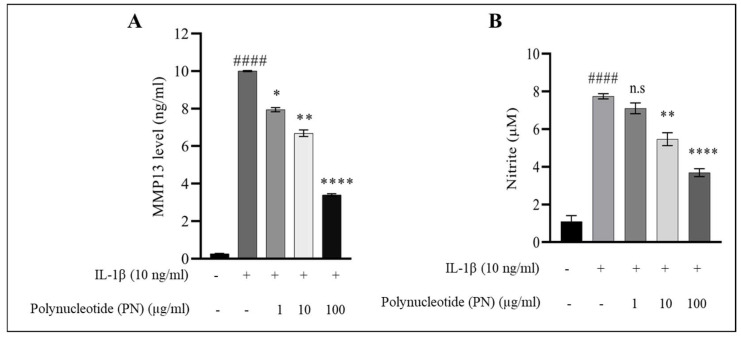
(**A**) Effect of polynucleotide (PN) treatment on the expression of MMP13 in IL−1β-induced HC-a. MMP13 levels were assessed using an enzyme-linked immunosorbent assay (ELISA) in HC-a treated with IL−1β and PN. (**B**) Effect of PN treatment on IL−1β-induced production of nitric oxide (NO) in HC-a chondrocytes. Nitric oxide levels were measured using a nitric oxide assay in HC-a chondrocytes treated with IL−1β and PN. Data are presented as the mean ± standard deviation (n = 3). Statistical analysis revealed significant differences, with the following symbols denoting significance values: #### *p* < 0.0001 compared to the control group (unstimulated cells); n.s (no significance); * *p* < 0.05, ** *p* < 0.01, and **** *p* < 0.0001 compared to the IL−1β-treated group.

**Figure 4 ijms-24-12282-f004:**
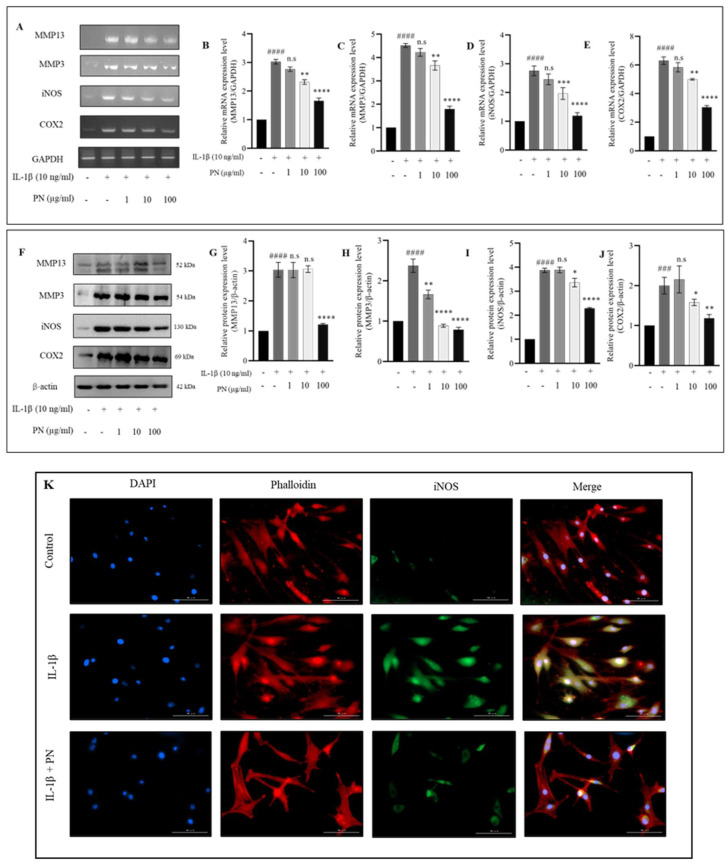
The effects of polynucleotide (PN) on inflammation and hypertrophy in IL−1β-induced HC-a. (**A**) Gene expression levels of MMP3, MMP13, iNOS, and COX2 were analyzed using RT-PCR. (**B**–**E**) Quantitative analysis of MMP3, MMP13, iNOS, and COX2 gene expression. (**F**) Protein expression levels of MMP3, MMP13, iNOS, and COX2 were determined through Western blot analysis. (**G–J**) Quantitative analysis of MMP3, MMP13, iNOS, and COX2 protein expression. (**K**) Immunofluorescence staining was performed to visualize the expression and localization of iNOS (bars = 100 μm; original magnification × 20). Data are presented as the mean ± standard deviation (n = 3). Statistical analysis revealed significant differences, with the following symbols denoting significance values: n.s (no significance); ### *p* < 0.001, #### *p* < 0.0001 compared to the control group (unstimulated cells); and * *p* < 0.05, ** *p* < 0.01, *** *p* < 0.001, and **** *p* < 0.0001 compared to the IL−1β-treated group.

**Figure 5 ijms-24-12282-f005:**
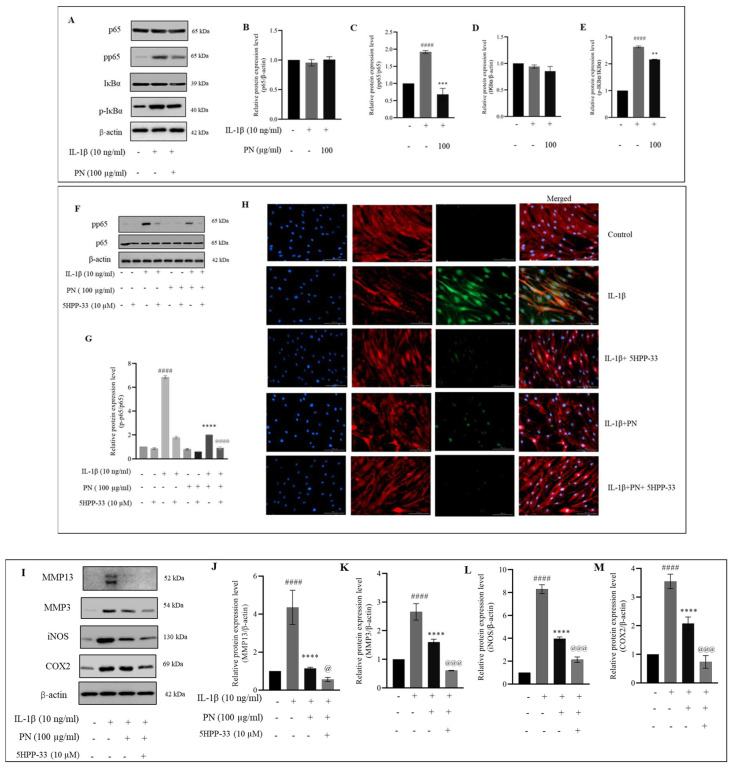
PN reduces IL−1β-induced HC-a inflammation through the NF-κB signaling pathway. (**A**) Whole cell lysate protein expression and (**B**–**E**) quantitative analyses of total and phosphorylated forms of p65 and IκBα protein expression levels. Effect of PN on the expression of pp65 protein in HC-a cells stimulated by IL−1β in the presence of the 5HPP-33 NF-κB signaling inhibitor, as evaluated through (**F**) protein expression levels, (**G**) quantitative analysis, and (**H**) immunofluorescence (bars = 100 μm; original magnification × 20). (**I**) Protein expression of pathway mediators (MMP3, MMP13, iNOS, and COX2) and (**J**–**M**) quantitative analysis of pathway mediators to determine the downstream effects of NF-κB inhibition with the 5HPP-33 NF-κB signaling inhibitor. The results of three independent experiments are presented as the mean ± standard deviation (SD). #### *p* < 0.0001 compared to the control group; ** *p* < 0.01, *** *p* < 0.001, and **** *p* < 0.0001 compared to the IL−1β-treated group; and @ *p* < 0.05, @@@ *p* < 0.001, and @@@@ *p* < 0.0001 compared to the PN treated group.

**Figure 6 ijms-24-12282-f006:**
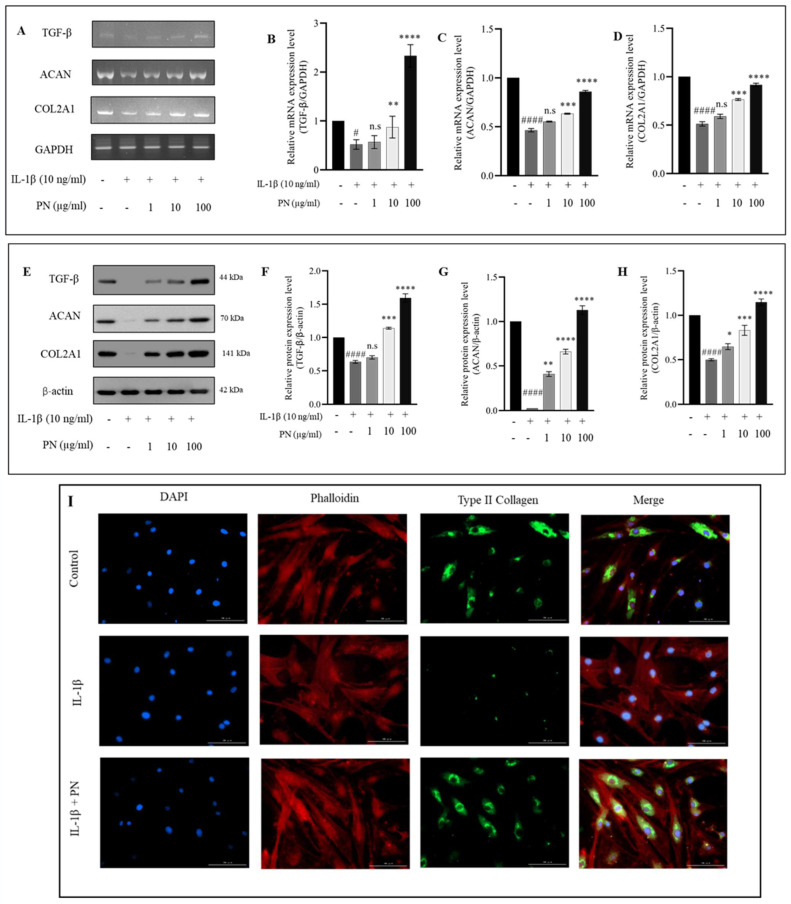
The effects of polynucleotide (PN) treatment on ECM synthesis in IL−1β-induced HC-a. (**A**) RT-PCR gene expression and (**B**–**D**) quantitative analysis of ECM components, including TGF-β, collagen type II (COL2A1), and aggrecan (ACAN). (**E**) Protein expression levels and (**F**–**H**) quantitative analysis of protein levels for TGF-β, COL2A1, and ACAN. (**I**) Immunofluorescence staining for visualization of ECM component, COL2A1, expression and localization (bars = 100 μm; original magnification × 20). Results are presented as mean ± standard deviation (SD) from three independent experiments. Statistical significance was determined as follows: # *p* < 0.05 and #### *p* < 0.0001 compared to the control group; n.s (no significance); * *p* < 0.05, ** *p* < 0.01, *** *p* < 0.001, and **** *p* < 0.0001 compared to the IL−1β-treated group.

**Figure 7 ijms-24-12282-f007:**
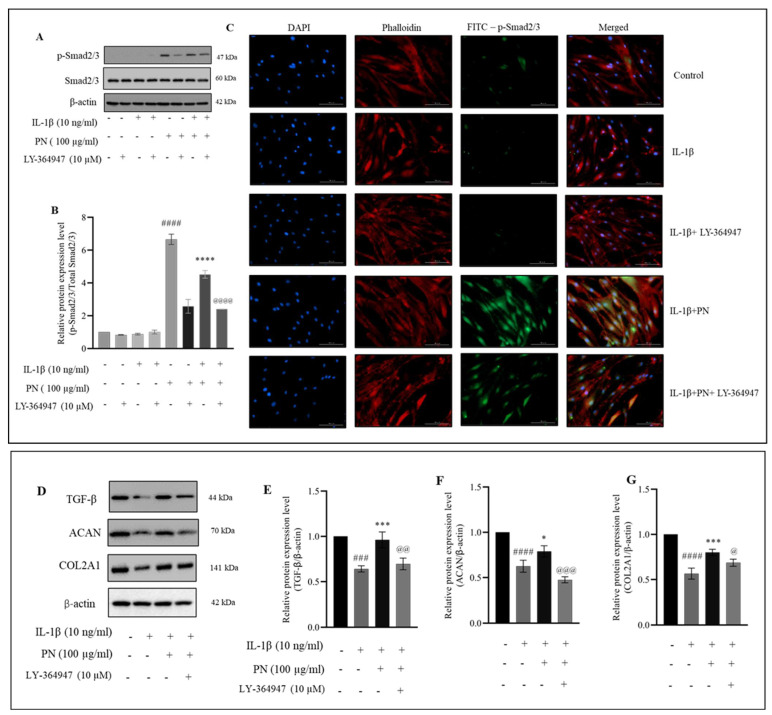
PN enhances Smad2/3 phosphorylation in HC-a. (**A**) Protein expression levels of p-Smad2/3 and (**B**) quantitative analysis, along with (**C**) immunofluorescence staining, demonstrate the impact of PN treatment on IL−1β-induced p-Smad2/3 protein expression in HC-a chondrocytes in the presence of the Smad2/3 inhibitor LY-364947 (bars = 100 μm; original magnification × 20). (**D**) Protein expression levels and (**E**–**G**) quantitative analysis of pathway mediators, including TGF-β, COL2A1, and ACAN, were assessed to investigate the downstream effects of Smad2/3 inhibition using the LY-364947 inhibitor. Results are presented as the mean ± standard deviation (SD) from three independent experiments. Statistical significance was determined as follows: ### *p* < 0.001, #### *p* < 0.0001 compared to the control group, * *p* < 0.05, *** *p* < 0.001, and **** *p* < 0.0001 compared to the IL−1β-treated group and @ *p* < 0.05, @@ *p* < 0.01, @@@ *p* < 0.001, and @@@@ *p* < 0.0001 compared to the PN treated group.

**Figure 8 ijms-24-12282-f008:**
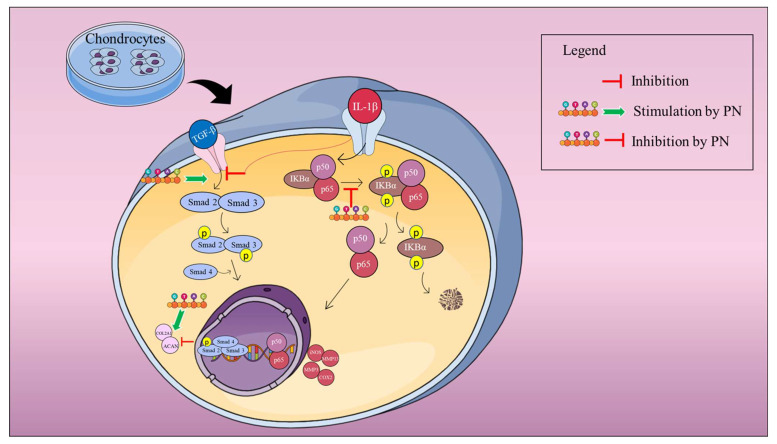
A schematic explanation of the mechanism by which PN inhibits the upregulation of p-NF-κB (pp65) expression in chondrocytes as a result of IL−1β stimulation. IL−1β alters the transcriptional activity of Smad2/3 by decreasing the expression of downstream mediators, such as COL2A1 and ACAN. The effect of PN is exerted through the reduction of p-NF-κB (pp65) expression and the elevation of Smad2/3 expression.

**Table 1 ijms-24-12282-t001:** List of primers used for experimental analysis.

Primer Name.	Forward Primer	Reverse Primer	Melting Temp
hCollagen II	TCTGCAACATGGAGACTGGC	GAAGCAGACCGGCCCTATGT	F 55.7 R 57.6
hAggrecan	ACGAGTGGCAGCGGTGAAT	GCCCTTCTCCTGCCTCTTG	F 57.6 R 56
hCOX2	TTC AAATGAGATTGTGGGAAA	AGATCATCTCTGCCTGAGTATCTT	F 51.9 R 52.0
hMMP3	GGCAGTTTGCTCAGCCTATC	GTCACCTCCAATCCAAGGAA	F 53.8 R 53.4
hMMP13	GATGAAGACCCCAACCCTAAA	CTGGCCAAAATGATTTCGTTA	F 54.3 R 54.1
hiNOS	AGCGGGATGACTTTCCAAGA	CTCCCGTCAGTTGGTAGGTT	F 59.02 R 59.03
hGAPDH	ACCACAGTCCATGCCATCAC	TCCACCACCCTGTTGCTGTA	F 55 R 55.9

## Data Availability

All data generated or analyzed during this study are included in this article.

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
