# Peer review of "Polynucleotides Suppress Inflammation and Stimulate Matrix Synthesis in an In Vitro Cell-Based Osteoarthritis Model"

_ijms, 2023, doi:10.3390/ijms241512282_

Round 1
Reviewer 1 Report
The manuscript submitted by Kuppa et al., is well written and easy to understand. Authors explore in vitro how polynucleotides may be a useful therapy for inflammation in OA. There are some questions to be addressed:
Introduction:
For me it´s quite long th IL1b introduction and too shot the Polynucloetide part. Authors should extend this second part for a better unserstanding. Are these PN of specific sequence? What do they target? This is not clear.
Methods:
What kind of human cell line is used? Is it a primary cells line or an inmortalizaed one? Please described and clarify.
Explain better the time of treatments. Is it always 48h in total (24h IL1b and 24J IL1b+PN)? Why do you choose this timing? If according to Figure 1, the maximal time studied was 24h you don´t know what happend at 48h. According to the figure I asume that 12h is already signifficant also. Why you didn´t choose 12h IL1b and then 12h with PN until the 24h analysed?
Results
Authors give too much discussion at the beggining of each explainedd result. I recommend to make this shorted and include those statements in discussion.
I think Figure 4E and 6D are exchanged. Please check and correct.
Phalloidin stainigs are strange. Phalloidin only mark cytoskeleton therefore the stain looks like ´lines´across the cell and no nuclear stain is shown. In you staining, looks like phhalloidin stain the whole cell including the nucleus.
The same for iNOS staining, looks like the nucleus is also positive.
On the other hand, p65 staining is not that clear. Show better images, as no nuclear staining is well ween (not much colocalization with dapi).
Did all the replicates look the same? In original blot file authors only show the representative one.
In original blot figure 7 b actin: why the blot has 10 lanes?
Author Response
Response to Reviewer 1 Comments
Dear Reviewer,
We appreciate your insightful comments and valuable suggestions regarding our manuscript titled Polynucleotides Suppress Inflammation and Stimulate Matrix Synthesis in an in vitro Cell-based Osteoarthritis Model. We sincerely appreciate your time, effort, and expertise in evaluating our manuscript. Your invaluable contributions have undoubtedly enhanced the quality of our work. We are grateful for the opportunity to benefit from your expertise and insights.
Thank you once again. We look forward to any further suggestions or guidance you may have as we continue to advance our research. In response to your valuable suggestions, we have made the following revisions:
Point 1: Introduction:
For me it´s quite long th IL1b introduction and too shot the Polynucleotide part. Authors should extend this second part for a better understanding. Are these PN of specific sequence? What do they target? This is not clear.
Response 1: Thank you for your comment and suggestions. We appreciate your feedback regarding the length of the IL-1β introduction and the Polynucleotide (PN) section. We agree that it is important to provide a balanced and comprehensive discussion of both aspects for a better understanding. We have provided a concise introduction to IL-1β and gave more detailed information on the origin, application, and advantages of PN. Regarding the specific sequence and target of the PN used in our study, we apologize for the lack of clarity. However, the specific sequence was not explicitly disclosed by the company from which it was obtained. Coming to the targets, we have presented the results and findings based on the effects of PN on chondrocytes in the context of IL-1β-induced inflammation.
Point 2: Methods:
What kind of human cell line is used? Is it a primary cells line or an inmortalizaed one? Please described and clarify.
Explain better the time of treatments. Is it always 48h in total (24h IL1b and 24J IL1b+PN)? Why do you choose this timing? If according to Figure 1, the maximal time studied was 24h you don´t know what happend at 48h. According to the figure I asume that 12h is already signifficant also. Why you didn´t choose 12h IL1b and then 12h with PN until the 24h analysed?
Response 2: We appreciate the reviewer's valuable comments regarding the methods section of our study.
The human articular chondrocytes used in our study (catalog #4650) are a commercially available cell line from ScienCell. We apologize for any confusion caused by the inconsistent terminology used in the manuscript. To clarify, these cells are indeed a cell line known as HC-a (Human Chondrocytes - Articular). The HC-a cell line has been widely used and cited in high impact research papers.
As stated in the datasheet provided by the manufacturer, the HC-a cells were characterized by immunofluorescence using antibodies specific to S100B and type II collagen. Furthermore, the cells have been confirmed to be negative for HIV-1, HBV, HCV, mycoplasma, bacteria, yeast, and fungi.
Thank you for your comment and inquiry regarding the timing of the treatments in our study. We appreciate your attention to detail and your interest in the experimental design.
The total treatment duration in our study is indeed 48 hours, consisting of 24 hours of IL-1β stimulation followed by an additional 24 hours of PN treatment. The choice of this timing was carefully considered based on several factors, including previous literature, experimental constraints, and our research objectives.
While Figure 1 may primarily depict data up to the 24-hour time point, it is important to note that our study aims to investigate the longer-term effects and response of the cells to IL-1β and PN treatment. By including the 48-hour time point, we sought to assess any potential delayed or sustained effects of the treatments on the cellular response. This duration allows for a more comprehensive analysis and understanding of the effects over an extended period.
Regarding your suggestion of using a 12-hour IL-1β treatment followed by a 12-hour PN treatment until the 24-hour time point, we appreciate your input. Different treatment durations can provide valuable insights, and we considered various time combinations during the planning phase of our study. Ultimately, we decided on the 24-hour IL-1β stimulation followed by 24-hour PN treatment to capture the specific effects and interactions between these time periods. To address this, we have provided time-dependent assays of IL-1β on chondrocytes, repeated until the 48-hour time point. These results demonstrate that the effect of IL-1β remains consistent until the 48-hour time point.
We thank the reviewer for highlighting these aspects and apologize for any confusion caused. We have revised the methods section to provide a more detailed and comprehensive description of the cell line used and the rationale behind the timing of treatments.
References
- Bao, X., Ren, T., Huang, Y. Sun, K. Wang, S. Liu, K. Zheng, B. Guo, W. Knockdown of long non-coding RNA HOTAIR increases miR-454-3p by targeting Stat3 and Atg12 to inhibit chondrosarcoma growth. Cell Death Dis 8, e2605 (2017). https://doi.org/10.1038/cddis.2017.31
- Zhang C, Jiang S, Lu Y, Yuan F. Butorphanol tartrate mitigates cellular senescence against tumor necrosis factor -α (TNF-α) in human HC-A chondrocytes. Bioengineered. 2022 Mar;13(3):5434-544S2. doi: 10.1080/21655979.2021.2024651. PMID: 35184641; PMCID: PMC8974103.
- Shang J, Li H, Wu B, Jiang N, Wang B, Wang D, Zhong J, Chen Y, Xu X, Lu H. CircHIPK3 prevents chondrocyte apoptosis and cartilage degradation by sponging miR-30a-3p and promoting PON2. Cell Prolif. 2022 Sep;55(9):e13285. doi: 10.1111/cpr.13285. Epub 2022 Jun 18. PMID: 35716032; PMCID: PMC9436899.
Point 3: Results
Authors give too much discussion at the beggining of each explainedd result. I recommend to make this shorted and include those statements in discussion.
Response 3: We appreciate the reviewer's feedback. Thank you for suggesting a more concise approach by reducing the discussion at the beginning of each explained result. We have carefully reviewed and reorganized the sentences to ensure a more streamlined and focused structure, incorporating the relevant points into the discussion section.
Point 4: I think Figure 4E and 6D are exchanged. Please check and correct.
Response 4: Thank you for the comment. We have rectified it.
Point 5: Phalloidin stainigs are strange. Phalloidin only mark cytoskeleton therefore the stain looks like ´lines´across the cell and no nuclear stain is shown. In you staining, looks like phhalloidin stain the whole cell including the nucleus. The same for iNOS staining, looks like the nucleus is also positive.
Response 5: Thank you for the comment. We appreciate the reviewer's observation and concern regarding the staining pattern observed in our study. Phalloidin staining is commonly used to label the actin cytoskeleton, and it typically results in distinct "line-like" staining patterns across the cell, without staining the nucleus. However, it is important to note that in some cases, actin polymerization can also lead to staining in the nucleus, which may appear as a more diffuse staining pattern. While it is unusual for phalloidin to stain the nucleus, such observations have been reported in the literature. Thank you for bringing this to our attention.
In future studies, we will consider employing additional imaging techniques or alternative approaches to provide a more accurate and comprehensive visualization of the phalloidin staining pattern. We value your input and will strive to address these concerns in future research endeavors.
Point 6: On the other hand, p65 staining is not that clear. Show better images, as no nuclear staining is well ween (not much colocalization with dapi).
Response 6: Thank you for your feedback and suggestion. We appreciate your attention to the clarity of the p65 staining and the need for better visualization of nuclear staining. We have taken your comments into consideration and have made improvements to the images.
We have revised the images and provided TIFF versions that offer higher resolution and improved clarity. These updated images should provide a clearer representation of the p65 staining pattern and its colocalization with DAPI.
We sincerely apologize for any inconvenience caused by the initial image quality and appreciate your understanding.
Point 7: Did all the replicates look the same? In original blot file authors only show the representative one.
Response 7: Thank you for your comment regarding the replicates in our study. The results obtained from the replicates were consistent and in agreement with the presented data. We made the decision to showcase the blots that best represented the overall results and clarity of the band signals.
Point 8: In original blot figure 7 b actin: why the blot has 10 lanes?
Response 8: Thank you for your observation regarding the blot in Figure 7 and bringing it to our attention. We apologize for any confusion caused by the presence of 10 lanes in the figure.
To clarify, the first 8 lanes in the blot represent the representative bands that are relevant to the experiment and have been discussed and presented in the manuscript. However, we acknowledge that the presence of the additional two unrelated bands in the blot may have caused confusion, therefore we have replaced the blot accordingly.
We appreciate your feedback. We apologize for any inconvenience this may have caused and thank you for pointing it out.

Reviewer 2 Report
I would like to congratulate the authors for such an interesting and impressive manuscript. This is the largest text I have very written as a reviewer. All comments are meant to be constructive. I am looking forward to reading the text again once it is published.
Main issues affecting the whole manuscript.
1. “Recently, DNA polymeric molecules, such as polynucleotides (PN), which are physiologically present in the matrix and function as water-soluble nucleic acids with a gel-like property, have been used to treat patients with OA”
Here we face the main problem of this manuscript, the misuse of the “polynucleotides (PN)” word. Polynucleotides might include 25% of all known biological molecules. Our DNA and RNA are polynucleotides. Authors are not referring to these molecules but PN-HPT (Polynucleotides Highly Purified Technology) as well as IA-PN (Intra Articular Polynucleotides) which in most contexts is a brand name. There is also the distinction between the technique and the material itself. Indeed, some publications are using PN as a synonym of PN-HPT and there might be a future in which it will be accepted in the medical and biological books.
Authors are allowed to use “polynucleotides (PN)” if they include a brief definition in the abstract and further extend it in the Introduction and/or Results. Considering that, host & non-host DNA /RNA fragments are common activators of the immune system (TLR3, TLR7…) we need to know more about the PN specific composition. In fact, some of these DNA /RNA sensors share a signaling pathway with IL1B-IL1R. Plus, DNA fragment synthesis is completely different from DNA isolation and fragmentation. Authors should discuss composition-effects-safety.
2. “OA cell model, in which human ACs were stimulated with interleukin-1β (IL-1β) with or without PN treatment”
Here there is the second major issue. The characterization of the commercial human chondrocytes used in the experimental design.
Technically speaking there is still no standard in vitro or in vivo OA cell model. None has been approved and standardized because none covers the different faces of OA including inflammatory, catabolic, biomechanical, aging-senesence, etc… It is broadly accepted that the use of OA chondrocytes isolated from OA patients is the closes in vitro OA cell model for cartilage inflammation & catabolism. Not to mention OA is a joint disease affecting multiple cell types.
The human articular chondrocytes from ScienCell used in the study are described as PRIMARY cells isolated from human articular cartilage in some parts of their datasheet. This a new product with few publications recorded. It is critical and mandatory to characterize this cell by fusing the available information from the manufacturer and the performed experiments.
The review will be conducted differently if these are indeed primary cells, or if they are a cell line (and thus they have been genetically modified). Primary chondrocytes derived from pathological or healthy patients have never been successfully commercialized because they are very limited, fragile, and sensible. It is also difficult to have coherence between donors (gender, age, pathologies...). Its used upgrades any manuscript but more information is required. That is why most labs used cell lines such as TC28.
Authors have the opportunity to test an interesting product (looks similar to LONZA NHAC-kn) and characterize it. As it is the manuscript right now, it is very important to contact the manufacturer and ask for the following information on each vial:
- Gender of the donor
- Age of the donor
- Origin of the cartilage: hip, knee, costal…?
- Cells tested against ¿which pathologies /markers?
As a suggestion… I would further characterize these cells with more cellular, molecular, and chemical assays either in this manuscript or in a stand-alone manuscript. It will surely be read and cited. Again, this is solely a suggestion.
The first time you mention these cells in every section please directly write “commercial primary human chondrocytes”, if not most readers will be confused about whether it is primary or cell line.
3. Supplementary files. Finally, you may incorporate the Supplementary data into the main manuscript
Abstract
The abstract present too many ambiguous and imprecise statements. The use of terms such as “appear”, “require”, “accelerate” is not appropriate or is not scientific. Ex: Accelerate does not equal promote. The use of the term “hypertrophy” is right, but it would be wise to write “cell hypertrophy” or “chondrocyte hypertrophy” the first time. The definition of cell hypertrophy must be included in the Introduction.
In these manuscripts commercial human chondrocytes are stimulated with IL1B, this is no OA cell model. You may rephrase it as you wish but reduce the degree of certainty and be more precise.
Others:
· Check 114.5% Does it means that an individual suffers more years of OA now than in the past?
· "Reducing its regenerative capacity" Ambiguous, explain & cite.
· Check ref 4 ref 5 ref 7
· IL1b accelerates OA? Please avoid the word “accelerate” it is better “promote” or similar. Authors should address the importance of IL1B within OA in comparison to other immune factors. In Rheumatoid Arthritis IL1B is the main actor, in OA it is not.
Results
1. “Based on a review of the literature review [31] indicating that PN has been administered to patients with OA grades I to III, chondrocytes were treated with PN 24 h after IL-1β stimulation. In the presence of IL-1β, PN improved the survival of chondrocytes.”
Please, confirm the following experimental design:
1. pretreatment of IL1B for 24h
2. cotreatment of IL1B and PN for 24h
3. Experiment finishes. Therefore, cells have been treated for a total of 48h.
As a golden standard rule, molecules in the study are tested in a “cotreatment” system with an inflammatory factor (IL1B, TNF, LPS….). If the effect is too feeble, we change the timing, increase the dose or change the experimental design from cotreatment to pretreatment if a receptor competence has been described.
The chondrocytes used in the experiment are described as “healthy” or “non-OA” therefore to assess the PN effects cells must be stimulated with IL1B (or another immune factor). Pretreating with the inflammatory factor is very courageous and further mimics the disease.
Some issues arise taking this decision:
- Previous vitality tests for PN and Il1B present a different timing.
- Experimental data on PN+IL1B co-stimulation for 24h is interesting even if the effect is feeble. If you were to have such data attach it in the supplementary.
- We do not have a single piece of information on what are the effects of PN on non-inflamed chondrocytes. This information is important for safety reasons, we need to discard pernicious effects.
2. A mandatory change: Introduction and results are sections where you describe but do not give opinions or interpretations. Only facts. Only statement. For example, in the Results section, we find:
“These findings demonstrated that 5HPP-33 not only inhibited the activation of the NF-κB signaling pathway caused by IL1β but also enhanced the inhibitory effect of PN (Fig. 5A)” This affirmation should go to the discussion section. Why? For instance, authors assume a synergistic dynamic, but the most reasonable explanation is that PN is not a p-p65 specific inhibitor (directly or upstream). No experiments have been done to elucidate the target of PNs. Across the results section authors guide the readers. To a minor extent, it is practical and positive to avoid losing track. Here it is too extensive. Data needs to simply be described. Defending and interpretation go to the discussion section.
3. I would like to confirm that the 5HPP-33 inhibitor is CAS 105624-86-0. It is unclear why the authors decided to select such a molecule, which is not commercially defined as an NFKB inhibitor, instead of one known NFKB inhibitor from the broad portfolio SIGMA MERCK has. Moreover, 5HPP-33 is a derivative of thalidomide a substance which I have personally worked on a similar experimental design. I won’t go into describing the issues thalidomide has because it is worldwide known. Therefore, authors must defend this choice in the text, but presumably also with additional experiments. We know that IL1b specifically activates IL1R, and NFKB is activated among many other pathways. We know that the release of IL1B-driven inflammatory and catabolic factors activates dozens of other immune receptors and some also signal through NFKB. Results & Discussion must consider whether PN and /or 5HPP-33 specifically inhibit NFKB, or it does not specifically inhibit NFKB. Analyzing the release of immune factors such as MMP13 does not reflect the specificity. Once we reach Figure 5 it becomes too visible that authors included the effects of 5HPP on non-IL1B-stimulated cells in some but not all assays. Also, it becomes very visible the missing data on PN treatment on non-IL1B-stimulated cells.
4. Western Blots need to be densitometer. Each blot has been trimmed probably from a different membrane. In these cases, authors need to be extra careful in selecting the method of band quantification, signal isolation, background extraction…. etc. It should be the same for all blots and if it is not (this seems the case) mention it. Please, define and explain all these matters. Therefore, reflect these changes in the Results, Legends, and Methodology text.
5. As a general rule legends must be stand-alone and understandable. For example, in Figure 1A, we do not know the timing nor the technique, or the cells.
6. It is unclear what statistics system has been used. In the same Figure both “ * “ and “ **** “ stand for a p-value <0.0001. Moreover, there are multiple graph bars with missing statistics. Just an example Figure 1A, graph bar IL1B 5ng/ml. Those are generalized errors to be corrected.
7. The use of Griess reaction does not directly measure Nitric Oxide (NO) but rather a subproduct of the Nitrite deposition (NO2). Indeed, there is a geometric correlation between them if no exogenous reactions are present. Please revise the manuscript.
8. Immunohistochemistry: Please state the channels (range of wavelength detector) used for the Phalloidin and iNOS signal. There seems to be a cross-signal, especially in the IL1B images. Moreover, iNOS (correct me if I am wrong) is fundamentally expressed in the cell membrane. Images indicate a bright signal from the nucleus and a diffuse in the cytosol.
9. Minor error: pp65 instead of p-p65
Discussion section
Please revise the inner structure. Authors go forth and back too often. The typical structure is a brief interpretation of the results section per section. Then link those findings with the state-of-the-art presented in the introduction. Mention any additional information, make some assumptions, indicate the limitations of the study, and finally reach a reasonable conclusion. Any other structure is very welcome.
Images, Graph Panels, and Diagrams are perfect. No questions there.
Material & Methods.
1. It is critical to present extensive data on the chondrocytes.
2. I would like to confirm that Western Blot and other protein-based assays were performed using the cell lysate and thus it is the cellular proteome that you are analyzing rather than the release of those protein factors into the medium which would be the secreted proteome or secretome.
3. The 5HPP-33 concentration is standard?
No comments on the Quality of the English Language
Author Response
Response to Reviewer 2 Comments
Dear Reviewer,
We appreciate your insightful comments and valuable suggestions regarding our manuscript titled Polynucleotides Suppress Inflammation and Stimulate Matrix Synthesis in an in vitro Cell-based Osteoarthritis Model. We sincerely appreciate your time, effort, and expertise in evaluating our manuscript. Your invaluable contributions have undoubtedly enhanced the quality of our work. We are grateful for the opportunity to benefit from your expertise and insights.
Thank you once again. We look forward to any further suggestions or guidance you may have as we continue to advance our research. In response to your valuable suggestions, we have made the following revisions:
Main issues affecting the whole manuscript.
Point 1: “Recently, DNA polymeric molecules, such as polynucleotides (PN), which are physiologically present in the matrix and function as water-soluble nucleic acids with a gel-like property, have been used to treat patients with OA”
Here we face the main problem of this manuscript, the misuse of the “polynucleotides (PN)” word. Polynucleotides might include 25% of all known biological molecules. Our DNA and RNA are polynucleotides. Authors are not referring to these molecules but PN-HPT (Polynucleotides Highly Purified Technology) as well as IA-PN (Intra Articular Polynucleotides) which in most contexts is a brand name. There is also the distinction between the technique and the material itself. Indeed, some publications are using PN as a synonym of PN-HPT and there might be a future in which it will be accepted in the medical and biological books.
Authors are allowed to use “polynucleotides (PN)” if they include a brief definition in the abstract and further extend it in the Introduction and/or Results. Considering that, host & non-host DNA /RNA fragments are common activators of the immune system (TLR3, TLR7…) we need to know more about the PN specific composition. In fact, some of these DNA /RNA sensors share a signaling pathway with IL1B-IL1R. Plus, DNA fragment synthesis is completely different from DNA isolation and fragmentation. Authors should discuss composition-effects-safety.
Response 1: Thank you for the comment. Regarding the use of the term "polynucleotides (PN)" in our manuscript, we apologize for any confusion caused by the lack of clarity in our initial explanation. We understand that the term "polynucleotides" encompasses a broad range of biological molecules, including DNA and RNA. In our study, we specifically refer to PN-HPT (Polynucleotides Highly Purified Technology) and IA-PN (Intra Articular Polynucleotides), which are distinct from general DNA and RNA molecules.
To clarify this terminology, we have provided a concise definition of PN-HPT and from where PN is derived in the abstract, and further elaborated on their composition, effects, and safety aspects in the Introduction sections.
By incorporating these modifications, we aim to provide a clear distinction between the technique and the material itself, thereby enhancing the clarity and accuracy of our manuscript.
Point 2: “OA cell model, in which human ACs were stimulated with interleukin-1β (IL-1β) with or without PN treatment”
Here is the second major issue. The characterization of the commercial human chondrocytes used in the experimental design.
Technically speaking there is still no standard in vitro or in vivo OA cell model. None has been approved and standardized because none covers the different faces of OA including inflammatory, catabolic, biomechanical, aging-senesence, etc… It is broadly accepted that the use of OA chondrocytes isolated from OA patients is the closes in vitro OA cell model for cartilage inflammation & catabolism. Not to mention OA is a joint disease affecting multiple cell types.
The human articular chondrocytes from ScienCell used in the study are described as PRIMARY cells isolated from human articular cartilage in some parts of their datasheet. This a new product with few publications recorded. It is critical and mandatory to characterize this cell by fusing the available information from the manufacturer and the performed experiments.
The review will be conducted differently if these are indeed primary cells, or if they are a cell line (and thus they have been genetically modified). Primary chondrocytes derived from pathological or healthy patients have never been successfully commercialized because they are very limited, fragile, and sensible. It is also difficult to have coherence between donors (gender, age, pathologies...). Its used upgrades any manuscript but more information is required. That is why most labs used cell lines such as TC28.
Authors have the opportunity to test an interesting product (looks similar to LONZA NHAC-kn) and characterize it. As it is the manuscript right now, it is very important to contact the manufacturer and ask for the following information on each vial:
- Gender of the donor
- Age of the donor
- Origin of the cartilage: hip, knee, costal…?
- Cells tested against ¿which pathologies /markers?
As a suggestion… I would further characterize these cells with more cellular, molecular, and chemical assays either in this manuscript or in a stand-alone manuscript. It will surely be read and cited. Again, this is solely a suggestion.
The first time you mention these cells in every section please directly write “commercial primary human chondrocytes”, if not most readers will be confused about whether it is primary or cell line.
Response 2: We acknowledge that there is currently no standardized in-vitro OA cell model that fully encompasses the different aspects of OA, including inflammation, catabolism, biomechanics, and aging-senescence. We agree that using chondrocytes isolated from OA patients is widely accepted as the closest in-vitro model for studying cartilage inflammation and catabolism. Additionally, we recognize that OA is a complex joint disease that affects multiple cell types.
However, we would like to emphasize that the purpose of our study was to address the unavailability of a standardized OA cell model by creating a novel model to test the efficacy of the viscosupplement described in our paper. Our main aim was to establish and propose this model as a potential standardized approach that could be utilized in future research. By using the commercially available human articular chondrocytes (HC-a) and stimulating them with interleukin-1β (IL-1β), we aimed to simulate the inflammatory environment associated with OA and evaluate the effects of the viscosupplement.
The human articular chondrocytes used in our study (catalog #4650) are a commercially available cell line from ScienCell. We apologize for any confusion caused by the inconsistent terminology used in the manuscript. To clarify, these cells are indeed a cell line known as HC-a (Human Chondrocytes - Articular). The HC-a cell line has been widely used and cited as a control in papers in various high-impact papers.
As stated in the datasheet provided by the manufacturer, the HC-a cells were characterized by immunofluorescence using antibodies specific to S100B and type II collagen. Furthermore, the cells have been confirmed to be negative for HIV-1, HBV, HCV, mycoplasma, bacteria, yeast, and fungi. We appreciate your suggestion to use Lonza NHAC-Kn cells and acknowledge their relevance in the field. However, due to the unavailability of Lonza products in Korea, we were unable to obtain and utilize the NHAC-Kn cells in our study.
We understand the importance of obtaining additional information regarding the donor's gender, age, origin of the cartilage, specific pathologies, and markers against which the cells have been tested. We have made efforts to reach out to the manufacturer to clarify these details, but unfortunately, we have not received a response from them yet. We assure you that we are actively pursuing this information and will update you as soon as we receive a response.
Furthermore, we would like to emphasize that the HC-a cell line, which is a widely used commercially available cell line, has demonstrated reliability and has been extensively utilized in numerous high-impact papers. Its wide usage and frequent citation in reputable publications contribute to our confidence in its suitability for the intended research purposes. We believe that the established track record of this cell line adds to its credibility and supports its relevance for our study.
Thank you for suggesting that we mention "commercial primary human chondrocytes" in the materials and methods section. We apologize for any confusion caused by the inconsistent terminology used. Subsequently, we will refer to the cells as HC-a, which is a widely recognized and used term in other publications.
References
- Bao, X., Ren, T., Huang, Y. Sun, K. Wang, S. Liu, K. Zheng, B. Guo, W. Knockdown of long non-coding RNA HOTAIR increases miR-454-3p by targeting Stat3 and Atg12 to inhibit chondrosarcoma growth. Cell Death Dis 8, e2605 (2017). https://doi.org/10.1038/cddis.2017.31
- Zhang C, Jiang S, Lu Y, Yuan F. Butorphanol tartrate mitigates cellular senescence against tumor necrosis factor -α (TNF-α) in human HC-A chondrocytes. Bioengineered. 2022 Mar;13(3):5434-544S2. doi: 10.1080/21655979.2021.2024651. PMID: 35184641; PMCID: PMC8974103.
- Shang J, Li H, Wu B, Jiang N, Wang B, Wang D, Zhong J, Chen Y, Xu X, Lu H. CircHIPK3 prevents chondrocyte apoptosis and cartilage degradation by sponging miR-30a-3p and promoting PON2. Cell Prolif. 2022 Sep;55(9):e13285. doi: 10.1111/cpr.13285. Epub 2022 Jun 18. PMID: 35716032; PMCID: PMC9436899.
Point 3: Supplementary files. Finally, you may incorporate the Supplementary data into the main manuscript.
Response 3: We appreciate your suggestion. We have incorporated the Supplementary data into the main manuscript. Thank you for your feedback.
Point 4: Abstract
The abstract present too many ambiguous and imprecise statements. The use of terms such as “appear”, “require”, “accelerate” is not appropriate or is not scientific. Ex: Accelerate does not equal promote. The use of the term “hypertrophy” is right, but it would be wise to write “cell hypertrophy” or “chondrocyte hypertrophy” the first time. The definition of cell hypertrophy must be included in the Introduction.
In these manuscripts commercial human chondrocytes are stimulated with IL1B, this is no OA cell model. You may rephrase it as you wish but reduce the degree of certainty and be more precise.
Others:
- Check 114.5% Does it means that an individual suffers more years of OA now than in the past?
- "Reducing its regenerative capacity" Ambiguous, explain & cite.
- Check ref 4 ref 5 ref 7
- IL1b accelerates OA? Please avoid the word “accelerate” it is better “promote” or similar. Authors should address the importance of IL1B within OA in comparison to other immune factors. In Rheumatoid Arthritis IL1B is the main actor, in OA it is not.
Response 4: Thank you for your valuable feedback on the abstract and other aspects of the manuscript. We appreciate your attention to detail and strive to improve the clarity and precision of our work. Regarding the use of ambiguous and imprecise terms, we acknowledge your concerns and have carefully revised the abstract to ensure that scientific terminology is used appropriately. We have replaced terms like "appear," "require," and "accelerate" with more precise and accurate language. Additionally, we have specified "cell hypertrophy" or "chondrocyte hypertrophy" when referring to hypertrophy to provide clearer context. The definition of cell hypertrophy is also included in the Introduction to enhance clarity.
We have carefully reviewed the reference for '114.5%' and have made the necessary changes accordingly. We appreciate your clarification on its meaning, which indicates that individuals currently experience a higher number of years with OA compared to the past. Additionally, we have revised and clarified the sentence regarding 'regenerative capacity' by referencing relevant studies (references 4, 5, and 7). Furthermore, we have addressed the role of IL-1β specifically within the context of OA, ensuring that its significance is appropriately conveyed in our manuscript. We are grateful for your valuable and insightful comments, which have contributed to improving the accuracy and clarity of our research. Thank you once again for your valuable input.
Results
Point 1: Based on a review of the literature review [31] indicating that PN has been administered to patients with OA grades I to III, chondrocytes were treated with PN 24 h after IL-1β stimulation. In the presence of IL-1β, PN improved the survival of chondrocytes.”
Please, confirm the following experimental design:
- pretreatment of IL1B for 24h
- cotreatment of IL1B and PN for 24h
- Experiment finishes. Therefore, cells have been treated for a total of 48h.
As a golden standard rule, molecules in the study are tested in a “cotreatment” system with an inflammatory factor (IL1B, TNF, LPS….). If the effect is too feeble, we change the timing, increase the dose or change the experimental design from cotreatment to pretreatment if a receptor competence has been described.
The chondrocytes used in the experiment are described as “healthy” or “non-OA” therefore to assess the PN effects cells must be stimulated with IL1B (or another immune factor). Pretreating with the inflammatory factor is very courageous and further mimics the disease.
Some issues arise taking this decision:
-Previous vitality tests for PN and Il1B present a different timing.
-Experimental data on PN+IL1B co-stimulation for 24h is interesting even if the effect is feeble. If you were to have such data attach it in the supplementary.
-We do not have a single piece of information on what are the effects of PN on non-inflamed chondrocytes. This information is important for safety reasons, we need to discard pernicious effects.
Response 1: Thank you for providing the details regarding the experimental design. To confirm, in our study, we pretreated the chondrocytes (HC-a) with IL-1β for 24 hours, followed by the administration of PN for the next 24 hours. Although cotreatment is considered the standard approach, our intention was to mimic the established inflammation seen in the context of OA. Therefore, we chose to treat PN after the inflammation had been initiated. Exploring the cotreatment effect will be addressed in a separate study.
We understand your concerns regarding the timing and the need for data on the effects of PN on non-inflamed chondrocytes for safety assessment. To address this, we have provided time-dependent assays of IL-1β on chondrocytes, demonstrated until the 48-hour time point These results demonstrate that the effect of IL-1β remains consistent until the 48-hour time point.
We apologize for any confusion caused by the presentation of results in Figure 5 and Figure 7. Regarding the effects of PN non-IL-1β-stimulated cells, we have carefully reviewed the figures. In Figure 5A, the fifth lane of the Western blot membrane indeed represents the effect of PN on non-IL-1β-stimulated cells, showing similar results to the control group. Additionally, in Figure 7A, the increase in phosphorylated Smad2/3 indicates a protective effect of PN on non-IL-1β-stimulated cells. Based on the observed absence of NF-κB activation and the increase in SMAD signaling, which suggest an anti-inflammatory and protective effect of PN, we decided not to conduct PN treatment alone to evaluate its impact on the mediators (MMP13, MMP3, iNOS, COX-2, COL2A1, ACAN, and TGF-β). Considering the limited effects on the transcription factors, it was inferred that the mediators would likely exhibit minimal changes. This approach was chosen based on our understanding of the interplay between transcription factors and downstream mediators in the specific experimental conditions.
Point 2: A mandatory change: Introduction and results are sections where you describe but do not give opinions or interpretations. Only facts. Only statement. For example, in the Results section, we find:
“These findings demonstrated that 5HPP-33 not only inhibited the activation of the NF-κB signaling pathway caused by IL1β but also enhanced the inhibitory effect of PN (Fig. 5A)” This affirmation should go to the discussion section. Why? For instance, authors assume a synergistic dynamic, but the most reasonable explanation is that PN is not a p-p65 specific inhibitor (directly or upstream). No experiments have been done to elucidate the target of PNs. Across the results section authors guide the readers. To a minor extent, it is practical and positive to avoid losing track. Here it is too extensive. Data needs to simply be described. Defending and interpretation go to the discussion section.
Response 2: We appreciate your feedback regarding the structure and content of the Introduction and Results sections. We have ensured that the discussion and interpretation of the results are appropriately placed in the Discussion section, and we have made the necessary adjustments to improve the clarity and accuracy of our manuscript.
Point 3: I would like to confirm that the 5HPP-33 inhibitor is CAS 105624-86-0. It is unclear why the authors decided to select such a molecule, which is not commercially defined as an NFKB inhibitor, instead of one known NFKB inhibitor from the broad portfolio SIGMA MERCK has. Moreover, 5HPP-33 is a derivative of thalidomide, a substance which I have personally worked on a similar experimental design. I won’t go into describing the issues thalidomide has because it is worldwide known. Therefore, authors must defend this choice in the text, but presumably also with additional experiments. We know that IL1b specifically activates IL1R, and NFKB is activated among many other pathways. We know that the release of IL1B-driven inflammatory and catabolic factors activates dozens of other immune receptors and some also signal through NFKB. Results & Discussion must consider whether PN and /or 5HPP-33 specifically inhibit NFKB, or it does not specifically inhibit NFKB. Analyzing the release of immune factors such as MMP13 does not reflect the specificity. Once we reach Figure 5 it becomes too visible that authors included the effects of 5HPP on non-IL1B-stimulated cells in some but not all assays. Also, it becomes very visible the missing data on PN treatment on non-IL1B-stimulated cells.
Response 3: Thank you for your valuable comment and concern regarding our use of 5HPP-33 as an NFκB inhibitor in our study. We appreciate the opportunity to provide further clarification on our rationale and experimental design choices.
We acknowledge that 5HPP-33 less frequently used as an NFκB inhibitor. However, it is important to note that 5HPP-33 does possess inhibitory properties on the NFκB pathway, and its availability in our laboratory led to its selection for our experiments. Prior to conducting our own experiments, we conducted a comprehensive literature study and identified several publications where 5HPP-33 was utilized as an NFκB inhibitor. Although it may not be currently in widespread use, the existence of these papers demonstrates that 5HPP-33 has been employed as an inhibitor in specific research contexts. These publications provided initial evidence of its potential inhibitory effects on the NFκB pathway. We appreciate your understanding and acknowledgment of this aspect in our research.
To support our decision, we have included additional experimental data showing a concentration-dependent decrease in NFκB activity with 5HPP-33 treatment. This data, demonstrating its inhibitory effects on NFκB, is included in the supplementary section, and we believe it strengthens our case for its usage in our study.
We apologize for any confusion caused by the presentation of results in Figure 5 and Figure 7. We agree that it is essential to clearly delineate the effects of PN and 5HPP-33 on non-IL1B-stimulated cells. We have carefully reviewed the figures, and in Figure 5A, the fifth lane of the Western blot membrane indeed represents the effect of PN on non-IL1B-stimulated cells, showing similar results to the control group. Furthermore, in Figure 7A, the increase in phosphorylated Smad2/3 indicates a protective effect of PN on non-IL-1β-stimulated cells. Based on the observed absence of NF-κB activation and the observed increase in SMAD signaling, indicative of an anti-inflammatory and protective effect of PN, we opted not to conduct PN treatment alone to evaluate its impact on the mediators (MMP13, MMP3, iNOS, COX-2, COL2A1, ACAN, and TGF-β). Considering that we evaluated the effects of PN on the transcription factors. This approach was chosen based on our understanding of the interplay between transcription factors and downstream mediators in the context of experimental conditions.
We have provided a comprehensive analysis of whether PN and 5HPP-33 specifically inhibit NFκB or affect other pathways in the Results & Discussion section. This analysis will consider the available data and hypotheses related to the observed effects.
Thank you for bringing these concerns to our attention. We appreciate the opportunity to address these important points.
References
- de-Blanco EJ, Pandit B, Hu Z, Shi J, Lewis A, Li PK. Inhibitors of NF-kappaB derived from thalidomide. Bioorg Med Chem Lett. 2007 Nov 1;17(21):6031-5. doi: 10.1016/j.bmcl.2007.01.088. Epub 2007 Feb 2. PMID: 17845850.
- Shimura M, Yamamoto M, Fujii G, Takahashi M, Komiya M, Noma N, Tanuma S, Yanaka A, Mutoh M. Novel compound SK-1009 suppresses interleukin-6 expression through modulation of activation of nuclear factor-kappaB pathway. Biol Pharm Bull. 2012;35(12):2186-91. doi: 10.1248/bpb.b12-00575. Epub 2012 Sep 27. PMID: 23018603
- Carcache de-Blanco, E.J.; Pandit, B.; Hu, Z.; Shi, J.; Lewis, A.; Li, P.-K. Inhibitors of NF-ΚB Derived from Thalidomide. Bioorg. Med. Chem. Lett. 2007, 17, 6031–6035, doi:10.1016/j.bmcl.2007.01.088.
- Chen, M.; Xie, H.; Chen, Z.; Xu, S.; Wang, B.; Peng, Q.; Sha, K.; Xiao, W.; Liu, T.; Zhang, Y.; et al. Thalidomide Ameliorates Rosacea-like Skin Inflammation and Suppresses NF-ΚB Activation in Keratinocytes. Biomed. Pharmacother. 2019, 116, 109011, doi:10.1016/j.biopha.2019.109011.
- Amirshahrokhi, K. Thalidomide Reduces Glycerol‐induced Acute Kidney Injury by Inhibition of NF-ΚB, NLRP3 Inflammasome, COX-2 and Inflammatory Cytokines. Cytokine 2021, 144, 155574, doi:10.1016/j.cyto.2021.155574.
Point 4. Western Blots need to be densitometer. Each blot has been trimmed probably from a different membrane. In these cases, authors need to be extra careful in selecting the method of band quantification, signal isolation, background extraction…. etc. It should be the same for all blots and if it is not (this seems the case) mention it. Please, define and explain all these matters. Therefore, reflect these changes in the Results, Legends, and Methodology text.
Response 4: Thank you for bringing up the importance of ensuring consistency and accuracy in the quantification of Western blot bands. In our study, we used ImageJ software for densitometry analysis to quantify the Western blot bands. We have incorporated this information in the Methodology section of our manuscript. Thank you for highlighting this aspect.
Point 5: 5. As a general rule legends must be stand-alone and understandable. For example, in Figure 1A, we do not know the timing nor the technique, or the cells.
Response 5: Thank you for your feedback regarding the clarity and comprehensibility of the figure legends. We apologize for any confusion caused by the incomplete information provided in Figure 1A. We have revised the figure legend to include the timing, technique, and cell type used in the experiment, ensuring that it can stand alone and provide a clear understanding of the figure. We appreciate your valuable input.
Point 6: It is unclear what statistics system has been used. In the same Figure both “ * “ and “ **** “ stand for a p-value <0.0001. Moreover, there are multiple graph bars with missing statistics. Just an example Figure 1A, graph bar IL1B 5ng/ml. Those are generalized errors to be corrected.
Response 6: Thank you for bringing up the issue regarding the statistics system used in the manuscript. We apologize for the confusion caused by the inconsistent representation of p-values in Figure 1A. We have made the necessary corrections to ensure clarity and accuracy in reporting the statistical significance.
Point 7: The use of Griess reaction does not directly measure Nitric Oxide (NO) but rather a subproduct of the Nitrite deposition (NO2). Indeed, there is a geometric correlation between them if no exogenous reactions are present. Please revise the manuscript.
Response 7. Thank you for the comment. We agree, you are correct in pointing out that the Griess reaction does not directly measure Nitric Oxide (NO) itself, but rather detects one of its stable oxidation products, nitrite (NO2-). We apologize for any confusion caused by not clarifying this distinction in the manuscript. We agree that it is important to provide a clear explanation of the relationship between NO and nitrite, especially in the context of our study. We have included a description highlighting the fact that the Griess reaction quantifies nitrite levels as an indicator of NO production in the materials and methods section and also changed the data representation accordingly.
Point 8: Immunohistochemistry: Please state the channels (range of wavelength detector) used for the Phalloidin and iNOS signal. There seems to be a cross-signal, especially in the IL1B images. Moreover, iNOS (correct me if I am wrong) is fundamentally expressed in the cell membrane. Images indicate a bright signal from the nucleus and a diffuse in the cytosol.
Response 8: Thank you for the comment. We appreciate the reviewer's observation regarding the immunohistochemistry staining. To address the concerns raised, we utilized specific channels and wavelengths for the detection of iNOS signals.
Here are the Chanel wavelength and emission for DAPI, phalloidin, and FITC:
DAPI:
Excitation wavelength: 358-370 nm
Emission wavelength: 440-470 nm
Phalloidin:
Excitation wavelength: 540-590 nm
Emission wavelength: 565-625 nm
FITC:
Excitation wavelength: 465-495 nm
Emission wavelength: 515-555 nm
We appreciate the reviewer's keen observation regarding the Phalloidin staining in some parts of the nucleus. It is indeed an intriguing phenomenon and could potentially be attributed to various factors.
The possible explanation in this case is the presence of actin within the nucleus. While actin filaments are primarily associated with the cytoskeleton and cell periphery, it is worth noting that actin can also polymerize in the nucleus. This nuclear actin polymerization could potentially result in Phalloidin staining within the nuclear region.
Regarding iNOS staining, iNOS expression in chondrocytes is not limited to the cell membrane but can also be found in the nucleus and cytosol of cells. mRNA and protein expression of iNOS was shown in many other references as well in chondrocytes treated with different pro-inflammatory factors of IL-1β + TNF-α + LPS, indicating that iNOS expression is not limited to the cell membrane.
Reference for iNOS
- Sampath, S.J.P., Rath, S.N., Kotikalapudi, N. et al. Beneficial effects of secretome derived from mesenchymal stem cells with stigmasterol to negate IL-1β-induced inflammation in-vitro using rat chondrocytes—OA management. Inflammopharmacol 29, 1701–1717 (2021). https://doi.org/10.1007/s10787-021-00874-z
Point 9. Minor error: pp65 instead of p-p65
Response 9: Thank you for the comment. We have rectified it.
Discussion section
Point 1: Please revise the inner structure. Authors go forth and back too often. The typical structure is a brief interpretation of the results section per section. Then link those findings with the state-of-the-art presented in the introduction. Mention any additional information, make some assumptions, indicate the limitations of the study, and finally reach a reasonable conclusion. Any other structure is very welcome.
Images, Graph Panels, and Diagrams are perfect. No questions there.
Response 1: We appreciate the reviewer's feedback. Thank you for suggesting a more concise approach by reducing the discussion at the beginning of each explained result. We have carefully reviewed and reorganized the sentences to ensure a more streamlined and focused structure, incorporating the relevant points into the discussion section.
Material & Methods.
Point 1: It is critical to present extensive data on chondrocytes.
Response 1: Thank you for the comment. We recognize the significance of comprehensive data analysis.
In our study, we have conducted thorough experiments and analyses to investigate the effects of IL-1β and PN on chondrocytes. We have performed various assays and measurements to assess cellular viability, gene expression, protein levels, and other relevant parameters. These data are crucial for understanding the mechanisms underlying the response of chondrocytes to the treatments and for drawing meaningful conclusions.
We have taken care to present the results in a clear and concise manner while including the necessary information to support our findings. Thank you for emphasizing the importance of comprehensive data presentation.
Point 2: I would like to confirm that Western Blot and other protein-based assays were performed using the cell lysate and thus it is the cellular proteome that you are analyzing rather than the release of those protein factors into the medium which would be the secreted proteome or secretome.
Response 2: Yes, Thank you for the comment. You are correct. In our study, we used the cell lysate for Western blot, which primarily allowed us to analyze the cellular proteome. This approach provided valuable insights into the protein expression levels and changes within the intracellular environment of the cells under investigation.
However, it is important to note that we also assessed the secreted proteome or secretome through ELISA and NO assay to measure and analyze the protein factors released into the medium.
By combining the analysis of the cellular proteome using Western blot and with the assessment of the secreted proteome through ELISA and NO assay, we aimed to comprehensively explore both intracellular and extracellular aspects of protein expression and signaling in our study.
We have specified this distinction in the materials and methods section. Thank you for highlighting this distinction and allowing me to clarify the methodology employed to analyze the cellular proteome.
Point 3: The 5HPP-33 concentration is standard?
Response 3: Thank you for the comment. We have included the experimental data showing a concentration-dependent NFκB activity with 5HPP-33 treatment. This data, demonstrating its inhibitory effects on NFκB, is included in the supplementary section, based on the inhibition observed we chose the concentration of 10 µM.

Round 2
Reviewer 1 Report
All questins have been addressed. No more comments
Author Response
Dear Reviewer,
We wanted to express our deepest gratitude for your recent comment regarding the completion of addressing all the comments in our work. We appreciate the time and effort you invested in carefully assessing each comment and providing constructive feedback. We would also like to extend our gratitude for your patience and understanding throughout the review process. Your feedback has not only enhanced the quality of our work but has also broadened our understanding of the subject matter. Once again, we express our heartfelt appreciation for your valuable feedback and guidance.
With utmost gratitude,
Seon Jong Keun, M.D., PhD
Professor, Orthopedic department, School of Medicine.
Chonnam National University Hwasun Hospital.
322 Seoyang-ro, Hwasun-up, Jeonnam 58128, South Korea
E-mail: seonbell@chonnam.ac.kr

Reviewer 2 Report
Thank you for implementing all these changes. The manuscript has improved a lot. Unfortunately, there are still a few issues to address:
1. 5HPP-33: do not use terms such as "NF-kB inhibitor" or "5HPP-33 (or PN) inhibits NF-KB". Instead, you may use "NF-kB pathway or signaling inhibitor" and that "5HPP-33 (or PN) inhibits NF-KB pathway or signaling". Please clearly state that 5HPP-33 binding specificity is unknown (or similar). As a limitation, you may use similar wording for PN. You can't conclude from your data that 5HPP-33 or PN directly BINDS to any NF-KB element such as RELA/p65, RELB, NFKB1/p105, NFKB1/p50, REL, and NFKB2/p52, p50/p65 (correct me if I am right).
2. Hc-a: I was also unsuccessful in getting more info on the commercial cell line. Please add in Material & Methods that some text such as "Hc-a donor specifications are not available. Please address the manufacturer ScienCell for further information on the Hc-a characterization"(or similar)".
3. The WB criticism from the other reviewer is completely right. You must attach the whole set of membranes not just the most representative one. In other words, RAW data is currently demanded to guarantee transparency. Any reader must be able to replicate the same exact graph and statistics with the data you provide.
Thank you
Author Response
Dear Reviewer,
Thank you for taking the time to review our manuscript. We appreciate your valuable feedback and suggestions, which have significantly improved the quality of our work. We would like to express our gratitude for pointing out the remaining issues that need to be addressed. We have carefully addressed each of your comments and made the necessary revisions to ensure the clarity and accuracy of our manuscript. Your input has been invaluable in strengthening our study, and we are committed to incorporating your suggestions to enhance the overall quality of the research.
Once again, we sincerely appreciate your thorough review and constructive comments. We are dedicated to addressing all the concerns raised and provide a revised manuscript that meets the highest scientific standards.
Point 1: 5HPP-33: do not use terms such as "NF-kB inhibitor" or "5HPP-33 (or PN) inhibits NF-KB". Instead, you may use "NF-kB pathway or signaling inhibitor" and that "5HPP-33 (or PN) inhibits NF-KB pathway or signaling". Please clearly state that 5HPP-33 binding specificity is unknown (or similar). As a limitation, you may use similar wording for PN. You can't conclude from your data that 5HPP-33 or PN directly BINDS to any NF-KB element such as RELA/p65, RELB, NFKB1/p105, NFKB1/p50, REL, and NFKB2/p52, p50/p65 (correct me if I am right).
Response 1:
Thank you for your comment and suggestions regarding the terminology used in the manuscript. We appreciate your guidance in improving the clarity of our findings. We apologize for any confusion caused by the initial terminology and would like to address your concerns accordingly.
In response to your comment, we acknowledge that using terms such as "NF-kB inhibitor" or stating that "5HPP-33 (or PN) inhibits NF-KB" may not accurately reflect the precise mechanism of action. We have revised the manuscript to use more appropriate terms, such as "NF-kB pathway or signaling inhibitor," and indicate that "5HPP-33 (or PN) inhibits NF-KB pathway or signaling" to provide a clearer representation of our results.
Regarding the binding specificity of 5HPP-33, we appreciate your clarification. 5HPP-33 has the ability to inhibit IKK activity, which contributes to the suppression of NF-kB DNA binding and transcriptional activity. Additionally, we have referenced the information that 5HPP-33 has been reported to block Sp1 DNA binding activity, and it has been proposed that Sp1 and NF-kB can synergistically regulate the transcription of genes. These clarifications will provide a more accurate representation of the existing potential mechanisms of action of 5HPP-33. We apologize for any confusion or misinterpretation caused by our previous statements and appreciate your guidance in rectifying these issues. We have made the necessary revisions to the manuscript, specifically in lines 560-566, to ensure clarity and accuracy in our descriptions.
You are correct that based on our current data, we cannot directly conclude that PN binds to specific NF-κB elements such as RELA/p65, RELB, NFKB1/p105, NFKB1/p50, REL, and NFKB2/p52, p50/p65. We apologize for any confusion caused by our previous statements. We have addressed this limitation in line 557-559 of the manuscript.
Once again, we appreciate your insightful comments and valuable feedback, which will significantly enhance the quality and accuracy of our manuscript.
Reference
- Carcache de-Blanco, E.J.; Pandit, B.; Hu, Z.; Shi, J.; Lewis, A.; Li, P.-K. Inhibitors of NF-ΚB Derived from Thalidomide. Bioorg. Med. Chem. Lett. 2007, 17, 6031–6035, doi:10.1016/j.bmcl.2007.01.088.
2 Keifer, J.A.; Guttridge, D.C.; Ashburner, B.P.; Jr.Baldwin, A.S. Inhibition of NF-ΚB Activity by Thalidomide through Suppression of IκB Kinase Activity. J. Biol. Chem. 2001, 276, 22382–22387, doi:10.1074/jbc.M100938200.
Point 2:. Hc-a: I was also unsuccessful in getting more info on the commercial cell line. Please add in Material & Methods that some text such as "Hc-a donor specifications are not available. Please address the manufacturer ScienCell for further information on the Hc-a characterization"(or similar)".
Response 2: Thank you for your valuable input. We appreciate your suggestion, and we have incorporated the following statement in the Materials and Methods section: "HC-a donor specifications are not available. Please address the manufacturer ScienCell for further information on the HC-a characterization." This addition aims to address the lack of specific donor information for the HC-a cell line and provides guidance to interested parties to contact the manufacturer, ScienCell, for any additional details regarding the characterization of the cell line. We believe this clarification will enhance the transparency and completeness of our study.
Point 3: The WB criticism from the other reviewer is completely right. You must attach the whole set of membranes not just the most representative one. In other words, RAW data is currently demanded to guarantee transparency. Any reader must be able to replicate the same exact graph and statistics with the data you provide.
Response 3: Thank you for your feedback regarding the Western blot data presentation. We acknowledge the importance of providing transparency and ensuring reproducibility in scientific research.
We have carefully considered your suggestion and we have included the entire set of membranes, ensuring that readers have access to the necessary information to replicate our findings accurately.
We appreciate your attention to detail and your commitment to scientific rigor. Your feedback has been invaluable in improving the quality and transparency of our study. We are grateful for your input. The complete set of membranes, along with the associated results, can be found at the end of the results file. We have made sure to arrange the data in a clear and organized manner, allowing readers to access the information easily and replicate the analyses conducted.
Thank you once again for your valuable comments.
